# The convergence rate of regularized learning in games: From bandits and uncertainty to optimism and beyond

**Angeliki Giannou**
Electrical and Computer Engineering
National Technical University of Athens
Athens, Greece
giannouangeliki@gmail.com

**Emmanouil V. Vlatakis-Gkaragkounis**
Department of Computer Science
Columbia University
New York, NY 10025
emvlatakis@cs.columbia.edu

**Panayotis Mertikopoulos**
Univ. Grenoble Alpes, CNRS, Inria, Grenoble INP, LIG &
Criteo AI Lab
panayotis.mertikopoulos@imag.fr

## Abstract

In this paper, we examine the convergence rate of a wide range of regularized methods for learning in games. To that end, we propose a unified algorithmic template that we call *"follow the generalized leader"* (FTGL), and which includes as special cases the canonical "follow the regularized leader" algorithm, its optimistic variants, extra-gradient schemes, and many others. The proposed framework is also sufficiently flexible to account for several different feedback models – from full information to bandit feedback. In this general setting, we show that FTGL algorithms converge locally to strict Nash equilibria at a rate which *does not depend* on the level of uncertainty faced by the players, but only on the geometry of the regularizer near the equilibrium. In particular, we show that algorithms based on entropic regularization – like the exponential weights algorithm – enjoy a linear convergence rate, while Euclidean projection methods converge to equilibrium in a *finite* number of iterations, even with bandit feedback.

## 1 Introduction

In the presence of uncertainty, the players of a game may not have full knowledge of its structure, "or the ability and inclination to go through any complex reasoning process to calculate an equilibrium. But the participants are still supposed to adapt by accumulating empirical information on the relative advantages of the various pure strategies at their disposal". This aphorism – originally due to Nash [34, p. 21] – constitutes the driving principle of game-theoretic learning, and highlights one of the field's most central questions: *Does learning with empirical observations lead to a Nash equilibrium? And, if so, at what rate?*

These questions have been at the forefront of game-theoretic research ever since the early days of the field, and they have recently received renewed attention via their connection to multi-agent reinforcement learning [43], generative adversarial networks [18], auctions [44], and many other applications where online decision-making plays a major role. Still, any attempt to provide a positive answer to these questions must wrestle with a major roadblock: the well-known impossibility result of Hart and Mas-Colell [19] shows that there are no uncoupled dynamics that converge to Nash equilibrium in *all* games, thus shattering any hope of obtaining a universal convergence result.

35th Conference on Neural Information Processing Systems (NeurIPS 2021), .

In view of the above, contemporary research on game-theoretic learning has focused on relaxing the feedback requirements of the players' learning processes, and understanding the stability – and *instability* – properties of different kinds of equilibria under popular learning algorithms. One property that stands out in this regard is the so-called "folk theorem" of evolutionary game theory [20], which can be stated as follows: Under the replicator dynamics – the continuous-time limit of the multiplicative / exponential weights (EW) algorithm [2, 29, 45] – *a Nash equilibrium is stable and attracting if and only if it is strict* (i.e., if and only if each player has a unique best response).

The replicator dynamics are the most widely studied model for evolution in population games, so the above equivalence essentially delineates what is and what isn't achievable in an evolutionary setting. In the context of online learning (our paper's main focus), a similar equivalence was obtained only recently [11, 15, 30], but it extends to the entire family of *"follow the regularized leader"* (FTRL) dynamics [41, 42], in both continuous [11, 30] and discrete time [15]. In particular, [15] studied the convergence of discrete-time FTRL models in the presence of uncertainty, and proved a high-probability, stochastic version of this equivalence that holds for several different types of feedback (full information, bandit, etc.). Thus, coupled with the prominence of FTRL in online and game-theoretic learning, strict Nash equilibria emerge as the only stable limit points of regularized learning under uncertainty.

**Our contributions.**   One important limitation of the above results is that they are qualitative in nature. Indeed, even though asymptotic stability guarantees that a learning process converges locally to a strict equilibrium, it provides no information about the *speed* of this convergence. In particular, especially for discrete-time models of regularized learning, asymptotic stability does not provide any guidance on how to tune the algorithm's hyperparameters (learning rate, mixing, etc.), and/or what to expect in terms of the number of iterations required to reach a neighborhood of a Nash equilibrium.

Our paper aims to provide quantitative answers to these questions for a wide array of regularized learning methods in the presence of uncertainty and limited information. To do so, we first introduce a flexible algorithmic framework – dubbed *"follow the generalized leader"* (FTGL) – that incorporates a broad spectrum of action choice mechanisms and feedback models. In more detail (and in analogy to FTRL), the FTGL template maintains a cumulative estimate for the payoff of each action available to the learner, and then selects a mixed strategy via a suitable "regularized" choice map. Specifically:

1. In terms of regularization, the FTGL template includes as special cases the standard logit choice and Euclidean projection methods (as well as all other standard regularizers used in practice).

2. In terms of the information used to update the "aggregate score" of each pure strategy, FTGL includes "vanilla" FTRL, its optimistic variants [10, 38–40], extra-gradient and mirror-prox methods [24, 25, 35], with either full, oracle-based, or bandit feedback.

In this general context, our main result may be summarized as follows. First, we introduce a "rate function" $\phi$ that depends *only* on the regularizer defining the learning process, and which captures the sensitivity of the induced choice map to external stimuli: for example, $\phi(x) = \exp(x)$ for entropic / logit choice models, whereas $\phi(x) = [x]_+$ for methods run with Euclidean projections. We then show that, with probability at least $1 - \delta$, the algorithm's local rate of convergence to a strict equilibrium $x^*$ is of the form $\|X_n - x^*\| \leq \phi(d - c \sum_{s=1}^n \gamma_s)$, where $\gamma_n$ is the method's learning rate and $c, d$ are constants with $c > 0$.

This result shows that the convergence speed of FTGL methods depends *only* on the choice of regularizer and learning rate: for example, EW methods run with a constant step size converge to an equilibrium at an exponential rate, whereas Euclidean regularization attains convergence in a *finite* number of iterations. From a regret-theoretic point of view, this is somewhat surprising because the regret guarantees of entropic FTRL (the EW algorithm) are far superior to those of FTRL with Euclidean regularization [5, 41].

Equally surprising is the fact that the type of feedback employed *does not affect the method's rate of convergence:* ceteris paribus, the base sequence of states generated by an FTGL method attains the *same* rate of convergence to strict Nash equilibria, whether run with full, partial, or bandit / payoff-based feedback. This comes into stark contrast with the corresponding rates of regret minimization, which depend crucially on the type of feedback received [6, 27]; in a certain, precise sense, this robustness in the face of uncertainty shows that regret minimization and convergence to Nash equilibrium are fundamentally different questions.

**Related work.** The convergence speed of methods based on the FTRL template – "vanilla", optimistic, or otherwise – have been studied extensively in the context of monotone games and variational inequalities; for a (highly incomplete) list of recent references, see [9, 10, 16, 17, 21, 23, 28, 31–33] and references therein. In this branch of the literature, there are two distinct threads: results concerning the convergence of the "time-average" of the process [16, 24, 33, 35], and those focusing on the algorithm's "last-iterate" [9, 10, 17, 21, 23, 28]. In the latter case (which is the one closest to our setting), the fastest achievable speed of convergence is exponential when the method is run with a finetuned constant step-size, perfect payoff gradient observations, and the operator defining the problem is strongly monotone and Lipschitz smooth. When run with stochastic gradients, the corresponding min-max optimal rate is $\mathcal{O}(1/T)$ under the same assumptions (zeroth-order rates are usually much worse). The apparent gulf between the rates of convergence obtained for monotone games and those obtained herein have to do with two crucial factors: first, we are studying *finite games*, which are typically not monotone; second, we are examining the algorithm's rate of convergence to *strict* equilibria, which are corner points of the problem's domain. This means that the geometry of the problem around a strict equilibrium is fundamentally sharper than the geometry around a solution of a generic monotone variational inequality, a fact which in turn explains the qualitatively different nature of the rates we obtain.

In the context of finite games, there have been several works examining the speed of convergence to the game's set of coarse correlated equilibria (CCE) by leveraging the algorithm's regret minimization properties, cf. [3, 4, 12, 13, 36, 44] and references therein. However, in addition to examining the algorithm's empirical average – as opposed to the induced day-to-day sequence of play – these results focus almost exclusively on CCE, which means that it is not possible to draw any conclusions about convergence to the game's Nash set – qualitatively or quantitatively. To the best of our knowledge, the closest work to our own in the literature is the paper of Cohen et al. [8] who showed that the EXP3 algorithm with explicit exploration converges at a sub-geometric rate in potential games; our analysis allows for a wider range of learning rates, so we are able to obtain faster convergence rates than Cohen et al. [8]. We are not aware of any other comparable results in the literature.

## 2 Preliminaries

**Finite games.** Throughout this work we consider $N$-players finite games in normal form. Formally, each *player*, indexed by $i \in \mathcal{N} = \{1, \ldots, N\}$, has a finite set of *pure strategies* $\alpha_i \in \mathcal{A}_i = \{1, \ldots, A_i\}$, and a *payoff function* $u_i \colon \mathcal{A} \to \mathbb{R}$, where $\mathcal{A} \coloneqq \prod_i \mathcal{A}_i$ is the space of all pure strategy profiles. For concision, we will denote such a game as a tuple $\Gamma = \Gamma(\mathcal{N}, \mathcal{A}, u)$.

During play, players can also play *mixed strategies*, i.e., probability distributions $x_i \in \mathcal{X}_i \coloneqq \Delta(\mathcal{A}_i)$ over their pure strategies. In this case, we will write $x_{i\alpha_i}$ for the probability that player $i \in \mathcal{N}$ selects $\alpha_i \in \mathcal{A}_i$ under $x_i$, $x = (x_1, \ldots, x_N)$ for the players' *mixed strategy profile*, and $\mathcal{X} \coloneqq \prod_i \mathcal{X}_i$ for the set thereof. Finally, when focusing on the mixed strategy of a particular player $i \in \mathcal{N}$, we will use the shorthand $(x_i; x_{-i}) \coloneqq (x_1, \ldots, x_i, \ldots, x_N)$ – and, similarly, $(\alpha_i; \alpha_{-i})$ for pure strategies.

Now, the expected payoff of player $i$ in a mixed strategy profile $x \in \mathcal{X}$ is given by

$$u_i(x) \equiv u_i(x_i; x_{-i}) = \sum_{\alpha_1 \in \mathcal{A}_1} \cdots \sum_{\alpha_N \in \mathcal{A}_N} u_i(\alpha_1, \ldots, \alpha_N) \cdot x_{1,\alpha_1} \cdots x_{N,\alpha_N} \tag{1}$$

where $u_i(\alpha_1, \ldots, \alpha_N)$ is the payoff of player $i$ in the action profile $\alpha = (\alpha_1, \ldots, \alpha_N) \in \mathcal{A}$. For posterity, we will also write $v_{i\alpha_i}(x) = u_i(\alpha_i; x_{-i})$ for the payoff that player $i$ would have gotten by playing $\alpha_i \in \mathcal{A}_i$ against the mixed strategy profile $x_{-i}$ of all other players. In this way, the *mixed payoff vector* of the $i$-th player can be seen as a vector field $v_i \colon \mathcal{X} \to \mathcal{Y}_i = \mathbb{R}^{\mathcal{A}_i}$ which can be written in components as

$$v_i(x) = (v_{i\alpha_i}(x))_{\alpha_i \in \mathcal{A}_i}. \tag{2}$$

Again, we will write $v(x) = (v_1(x), \ldots, v_N(x))$ for the ensemble of payoff vectors of all players and $\mathcal{Y} = \prod_i \mathcal{Y}_i$ for the space of payoff vectors respectively. Finally, in a slight abuse of notation, we will identify $\alpha_i$ with the mixed strategy that assigns all probability to $\alpha_i$, and we will write $v_i(\alpha) = (u_i(\alpha_i; \alpha_{-i}))_{\alpha_i \in \mathcal{A}_i}$ for the corresponding *pure payoff vector*.

**Nash equilibrium.** The most widely used solution concept in game theory is that of a Nash equilibrium i.e., a (possibly) mixed strategy profile $x^* \in \mathcal{X}$ that discourages unilateral deviations;

formally, $x^* \in \mathcal{X}$ is said to be a *Nash equilibrium* of $\Gamma$ if

$$u_i(x^*) \geq u_i(x_i; x^*_{-i}) \quad \text{for all } x_i \in \mathcal{X}_i \text{ and all } i \in \mathcal{N}. \tag{NE}$$

The set of pure strategies supported at the equilibrium component $x^*_i \in \mathcal{X}_i$ of each player will be denoted by $\text{supp}(x^*_i) = \{\alpha_i \in \mathcal{A}_i : x^*_{i\alpha_i} > 0\}$. In turn, the size of the support of $x^*$ leads to the following dichotomy: $x^*$ is called *pure* if $\text{supp}(x^*_i) \equiv \prod_{i \in N} \text{supp}(x^*_i)$ is a singleton and *mixed* otherwise.

Finally, we will also say that a Nash equilibrium $x^*$ is *strict* if (NE) holds as a *strict* inequality whenever $x_i \neq x^*_i$; obviously, strict equilibria are also pure, but the converse need not hold. Strict Nash equilibria play a key role in game theory because any unilateral deviation incurs a strict loss to the deviating player; put differently, if $x^*$ is strict, every player has a unique best response. In addition, they are the only equilibria that remain invariant under small generic perturbations of the game [14]; these robustness properties of strict equilibria will play a key role in the sequel.

## 3  Regularized learning

Throughout our paper, we will focus on a wide family of learning schemes that unfold as follows: At each stage $n = 1, 2, \ldots$, every player maintains a "score vector" $Y_{i,n} \in \mathcal{Y}_i$ whose components indicate the player's propensity to play a given pure strategy. More formally, this is captured by a player-specific *"regularized choice" map* $Q_i \colon \mathcal{Y}_i \to \mathcal{X}_i$ which outputs the player's mixed strategy $X_{i,n} = Q_i(Y_{i,n})$ as a function of $Y_{i,n}$ (see below for a detailed definition). Then, after selecting their actions and collecting their rewards, players also receive – or otherwise construct – an estimate $V_{i,n}$ of their mixed payoff vectors, which is used to increment their score variables and continue playing.

Formally, this learning process, which we call *"follow the generalized leader"* (FTGL), can be described via the round-by-round recursive rule

$$\begin{aligned} X_{i,n} &= Q_i(Y_{i,n}) \\ Y_{i,n+1} &= Y_{i,n} + \gamma_n V_{i,n} \end{aligned} \tag{FTGL}$$

where $\gamma_n > 0$ is a "learning rate" parameter such that $\sum_n \gamma_n = \infty$. The terminology FTGL alludes to the widely known "follow the regularized leader" algorithm, which is, historically speaking, the parent-scheme of FTGL. The link to regularization is provided by the method's choice map, which we detail below; the assumptions for the signal sequence $V_{i,n}$ are provided right after.

**3.1. The choice map.** The guiding principle behind the definition of the players' choice maps $Q_i \colon \mathcal{Y}_i \to \mathcal{X}_i, i \in \mathcal{N}$, as follows: Because the players' score variables $Y_{i,n}$ are assumed to represent an estimate of each strategy's cumulative payoff over time, $Q_i$ is defined as a "regularized" version of the best-response correspondence $y_i \mapsto \arg\max_{x_i \in \mathcal{X}_i} \langle y_i, x_i \rangle$.[1] On that account, we will consider *regularized best responses* of the general form

$$Q_i(y_i) = \arg\max_{x_i \in \mathcal{X}_i} \{\langle y_i, x_i \rangle - h_i(x_i)\} \tag{3}$$

where $h_i \colon \mathcal{X}_i \to \mathbb{R}$ denotes the $i$-th player's *regularization function*.

For concreteness, we will focus on a class of decomposable regularizers of the form $h_i(x_i) = \sum_{\alpha_i \in \mathcal{A}_i} \theta_i(x_i)$ where the so-called "kernel function" $\theta_i \colon [0, 1] \to \mathbb{R}$ is assumed continuous on $[0, 1]$, twice differentiable on $(0, 1]$, and strongly convex, i.e., $\inf_{(0,1]} \theta''_i > 0$. Of course, different regularizers give rise to different instances of (FTGL); two of the most widely used and cited examples are as follows:

**Example 3.1** (Entropic regularization and multiplicative/exponential weights)**.** Perhaps the most common representative of regularization functions is given by the entropic kernel $\theta(x) = x \log x$ i.e., $h(x_i) = \sum_{\alpha_i \in \mathcal{A}_i} x_{i\alpha_i} \log x_{i\alpha_i}$. This choice of regularizer is well-known to provide the *logit choice map* $\Lambda_i(y_i) = (\exp(y_{i\alpha_i}))_{\alpha_i \in \mathcal{A}_i} / \sum_{\alpha_i \in \mathcal{A}_i} \exp(y_{i\alpha_i})$. The resulting algorithm is known in the literature as the multiplicative/exponential weights algorithm [1, 2, 29, 41, 45].

**Example 3.2** (Euclidean projection)**.** Another popular regularizer is the quadratic penalty $h(x) = \sum_a x_a^2 / 2$, which yields the *payoff projection* map $\Pi(y) = \arg\min_{x \in \Delta} \|y - x\|^2$, cf. [26, 46].

---

[1] In this context, regularization can be seen as a means to reinforce exploration so as to avoid committing prematurely to a given strategy.

*Remark* 3.1. Examples 3.1 and 3.2 are archetypes of a fundamental dichotomy between regularization functions: in the former case, we have $\theta'(0) = -\infty$, so $h$ becomes *steep* at the boundary of the player's strategy space; in the later case, $\theta$ is differentiable at 0, so $h$ is non-steep. We will see that this steep/non-steep dichotomy has a crucial impact on the method's rate of convergence.

**3.2. The feedback model.** As we mentioned in the beginning of the section, the "payoff signal" $V_n$ contains information about the structure of the algorithm as well as the setting under consideration. Thus to account for as broad a range of algorithms as possible, we will assume that the players' signal sequence is of the general form

$$V_n = v(X_n) + Z_n \tag{4}$$

for some abstract error process $Z_n = (Z_{i,n})_{i \in \mathcal{N}}$. Tp be clear though, we should stress that *we do not assume* that $V_n$ is directly correlated to – or obtained by – the chosen mixed strategy $X_n$; this will be made clear in the range of models we present below.

To distinguish between random (zero-mean) and systematic (non-zero-mean) errors, we will further decompose $Z_n$ as $Z_n = U_n + b_n$, where

$$b_n = \mathbb{E}[Z_n \mid \mathcal{F}_n] \quad \text{and} \quad \mathbb{E}[U_n \mid \mathcal{F}_n] = 0 \tag{5}$$

with $\mathcal{F}_n$ denoting the history of $X_n$ up to stage $n$ (inclusive). Notice that, since the feedback signal is generated only *after* the player chooses a strategy, $V_n$ is not $\mathcal{F}_n$-measurable in general. On this account, we will make the following blanket assumptions for the input signal sequence $V_n$:

1. *Vanishing bias:*     $b_n$ converges uniformly to 0 as $n \to \infty$.           (A1)

2. *Bounded variance:*    $\mathbb{E}[\|U_n\|_*^q \mid \mathcal{F}_n] \leq \sigma_n^q$ for some $q > 2$.     (A2)

In the above $\sigma_n$ is assumed to be a deterministic, stage-specific, and possibly increasing bound on the variance of the noise component $U_n$; our precise assumptions for its growth (relative to $b_n$ or otherwise) will be detailed later in this section.

**Specific models.** So far, the formulation of (FTGL) has been kept intentionally abstract and devoid of any modeling assumptions for how the players' payoff signals are generated or estimated. To illustrate the width and breadth of (FTGL), we present of series of specific models below, including the popular FTRL and optimistic FTRL methods:

**Model 1** (FTRL with oracle-based feedback). Assume that each player chooses an action based on a given mixed strategy, and once every player has chosen an action, an oracle reveals to each player their corresponding pure payoff vector. Formally, at each round $n = 1, 2, \ldots$, each player chooses a pure strategy $\alpha_{i,n} \in \mathcal{A}_i$ based on a mixed strategy $X_{i,n} \in \mathcal{X}_i$ and subsequently observes the payoff vector

$$V_{i,n} = v_i(\alpha_n) = (u_i(\alpha_i; \alpha_{-i,n}))_{\alpha_i \in \mathcal{A}_i}. \tag{6}$$

Thus, in this case, (FTGL) boils down to the standard *"follow the regularized leader"* (FTRL) algorithm of [41, 42]. As for our basic feedback assumptions, we readily see that $b_{i,n} = 0$ and $U_{i,n} = v_i(\alpha_n) - v_i(X_n)$; hence:

- (A1) is trivially satisfied since $b_{i,n} = 0$.
- (A2) is again satisfied because $\|U_{i,n}\|_* = \|v_i(\alpha_n) - v_i(X_n)\|_* \leq 2 \max_{\alpha \in \mathcal{A}} \|v_i(\alpha)\|_*$, so $U_n$ has uniformly bounded moments for all $q \in [1, \infty]$.          §

**Model 2** (FTRL with bandit feedback). If the players only observe their realized rewards, they have to *construct* a model for $V_n$ based on incomplete information. This is the standard setting for multi-armed bandits [5, 6, 27], so it is often referred to as the "bandit feedback" model. In this case, the players' action selection process is as in Model 1 above, but the feedback signal sequence $V_n$ is now reconstructed by means of the importance-weighted estimator

$$V_{i\alpha_i,n} = \frac{\mathbb{1}\{\alpha_{i,n} = \alpha_i\}}{\hat{X}_{i\alpha_{i,n}}} u_i(\alpha_n) \tag{IWE}$$

where $\hat{X}_{i,n} = (1 - \varepsilon_n)X_{i,n} + \varepsilon_n/|\mathcal{A}_i|$ is the mixed strategy of the $i$-th player at stage $n$. Compared to $X_{i,n}$ the player's actual sampling strategy is now recalibrated by an *explicit exploration* parameter $\varepsilon_n \to 0$ whose role is to stabilize the learning process. The idea behind this adjustment is that even if a strategy has zero probability to be chosen under $X_n$, it will still be sampled with positive probability thanks to the mixing factor $\varepsilon_n$.

| Feedback | FTRL | OptFTRL | EG/MP |
|---|---|---|---|
| Full information | $b_n = 0$ 
 $M_n = 0$ | $\|b_n\|_* = \mathcal{O}(\gamma_n)$ 
 $M_n = 0$ | $\|b_n\|_* = \mathcal{O}(\gamma_n)$ 
 $M_n = 0$ |
| Oracle-based | $b_n = 0$ 
 $M_n = \mathcal{O}(1)$ | $\|b_n\|_* = \mathcal{O}(\gamma_n)$ 
 $M_n = \mathcal{O}(1)$ | $\|b_n\|_* = \mathcal{O}(\gamma_n)$ 
 $M_n = \mathcal{O}(1)$ |
| Bandit 
 (payoff-based) | $\|b_n\|_* = \mathcal{O}(\varepsilon_n)$ 
 $M_n = \Theta(1/\varepsilon_n)$ | $\|b_n\|_* = \mathcal{O}(\varepsilon_n)$ 
 $M_n = \Theta(1/\varepsilon_n)$ | $\|b_n\|_* = \mathcal{O}(\varepsilon_n)$ 
 $M_n = \Theta(1/\varepsilon_n)$ |

**Table 1:** Recasting different online learning algorithms within the general template of (FTGL).

The importance-weighted estimator (IWE) estimator may be seen as a special case of the model (4) with $U_{i,n} = V_{i,n} - v_i(\hat{X}_n)$ and $b_{i,n} = v_i(\hat{X}_n) - v_i(X_n)$. Both assumptions (A1),(A2) are again satisfied; indeed:

- For (A1): A standard calculation performed in **??** reveals that $\|b_{i,n}\|_* = O(\varepsilon_n)$. Thus our assumption is satisfied since $\varepsilon_n \to 0$.
- For (A2): Again a standard calculation presented in **??** reveals that $\|V_{i,n} - v_i(\hat{X}_n)\|_* = O(1/\varepsilon_n)$ and thus the noise has bounded moments, $\sigma_n = \Theta(1/\varepsilon_n)$ for all $q \in [1, \infty]$.

**Model 3** (OptFTRL with oracle-based feedback). Following Rakhlin and Sridharan [40], the so-called "optimistic" variant of FTRL is given by the recursive formula:

$$\tilde{Y}_{i,n} = Y_{i,n} + \gamma_n V_{i,n-1} \qquad \tilde{X}_{i,n} = Q_i(\tilde{Y}_{i,n}) \qquad Y_{i,n+1} = Y_{i,n} + \gamma_n V_{i,n} \qquad \text{(OptFTRL)}$$

In the above the payoff signal $V_{i,n}$, which depends on the state $\tilde{X}_n$, is generated as follows: at each round $n = 1, 2, \ldots$, every player $i \in \mathcal{N}$ picks an action $\alpha_{i,n} \in \mathcal{A}_i$ based on $\tilde{X}_{i,n} \in \mathcal{X}_i$ and observes the pure payoff vector $v_i(\alpha_n) \equiv (u_i(\alpha_i; \alpha_{-i,n}))_{\alpha_i \in \mathcal{A}_i}$. At first glance, it seems difficult to reconcile the above update structure with that of (FTGL); however, it is indeed possible to integrate (OptFTRL) within (FTGL) by considering the auxiliary states $X_n = Q(Y_n)$ (which are never played and are only used here for the analysis).

Indeed, each player's input signal is a special case of (4) with payoff feedback $V_{i,n} = v_i(\alpha_n)$, zero-mean noise $U_{i,n} = v_i(\alpha_n) - v_i(\tilde{X}_n)$ and bias $b_{i,n} = v_i(\tilde{X}_n) - v_i(X_n)$ that satisfy both the assumptions (A1),(A2). In more detail, we have:

- For (A1): $\|b_{i,n}\|_* = \|v_i(\tilde{X}_n) - v_i(X_n)\|_* = O(\gamma_n)$, which goes uniformly to 0 whenever $\gamma_n \to 0$.
- For (A2): $\|U_{i,n}\|_* = \|v_i(\alpha_n) - v_i(\tilde{X}_n)\|_* \le 2\max_{\alpha \in \mathcal{A}}\|v_i(\alpha)\|_*$ and thus the noise has bounded moments for all $q \in [1, \infty]$.

*Remark* 3.2. Based on the structure of the aforementioned algorithms, it is easy to check that we capture *a-fortiori* the model of a full-information payoff signal; for a more complete account of the different algorithms and feedback models see Table 1.

## 4 Analysis & Results

We are now in a position to state our main convergence results for (FTGL). We begin with a precise statement and discussion in Section 4.1; subsequently, we present the main proof techniques in Section 4.2.

**4.1. Statement and discussion of our main results.** Our analysis will focus exclusively on strict Nash equilibria. As we discussed in the introduction, the reason for this is that only strict Nash equilibria can be asymptotically stable under FTRL [11, 15], so they are the only reasonable candidates to consider when examining the rate of convergence of a regularized learning algorithm.[2]

---

[2]As a sidenote, we should remark here that FTGL also contains the optimistic FTRL algorithm, which *does* converge to mixed Nash equilibria in bilinear zero-sum games with *perfect, deterministic* feedback [16, 25, 32]. At first glance, this would seem to contradict the results of [11, 15], but one needs to bear in mind that the convergence of (OptFTRL) to mixed equilibria only occurs in settings with *perfect information* (i.e., $V_n = v(X_n)$ for all $n = 1, 2, \ldots$). In the presence of uncertainty, this convergence is destroyed [7, 22], so there is no contradiction with the results of [15]. Because we are primarily interested in learning with limited information and/or under uncertainty, we will not treat this somewhat fragile case.

To proceed, we will need one technical assumption linking the learning rate of (FTGL) and the bias/variance parameters of the driving feedback sequence $V_n$. This is as follows:

$$\text{The sequence } \delta_n := \frac{\sum_{k=1}^{n} \gamma_k^{1+\frac{q}{2}} \sigma_k^q}{\left[\sum_{k=1}^{n} \gamma_k\right]^{1+\beta q/2}} \text{ is summable for some } \beta < 1. \tag{A3}$$

Assumption (A3) imposes a growth condition on the method's learning rate relative to the bias and variance parameters of the input signal sequence $V_n$, but it is otherwise a technical prerequisite for the analysis to come. What is more important for our purposes is that the concrete models that we discussed in the previous section satisfy it for a wide range of the player-chosen parameters $\gamma_n$ (and $\varepsilon_n$ in the case of bandit-based algorithms); to streamline our presentation, we postpone a more precise discussion of this issue until after the statement of our main results.

The last element that we need to introduce concerns the players' choice of regularizer: clearly, since propensities are transformed to strategies via each player's individual choice map $Q_i: \mathcal{Y}_i \to \mathcal{X}_i$, it stands to reason that the underlying regularization function $h$ plays a major role in the method's rate of convergence. Indeed, given an update of the form $Y_{n+1} \leftarrow Y_n + \gamma_n V_n$, the method's strategy variable will move forward as $X_{n+1} \leftarrow X_n + \gamma_n JQ(V)V_n + \mathcal{O}(\gamma_n^2)$, where $JQ$ denotes the Jacobian matrix of $Q$. Thus, to leading order in $\gamma_n$, the update $X_{n+1} \leftarrow X_n$ is dominated by the first derivatives of $Q$.

By a relatively straightforward application of the Legendre identity from convex analysis ($Q = (\partial h)^{-1}$ in our context; see below for a precise statement), this rate of change is related to the inverse mapping of the derivative each $\theta_i$ (the kernel of the underlying regularizer). Motivated by this observation, we introduce below the algorithm's so-called *rate function*:

$$\phi_i(t) = \begin{cases} (\theta_i')^{-1}(t) & \text{if } t > \theta_i'(0^+), \\ 0 & \text{otherwise.} \end{cases} \tag{7}$$

The definition of the rate function $\phi$ captures the sensitivity of the choice map $Q$ in a very precise way: If the score difference corresponding to two pure strategies $\alpha, \beta \in \mathcal{A}_i$ grows as $y_\beta - y_\alpha = t$ for some $t > 0$, then the probability of playing the strategy with the lesser score must be less than the probabiity of playing the strategy with the higher score. The precise amount of this disparity of course depends on the player's choice function $Q$ and $\phi$ acts as a "transfer" function in this regard. Specifically, as we show in detail later, we have $x_\alpha = \phi(-\Theta(t))$, i.e., $\phi$ determines the rate at which $x_\alpha$ vanishes. For different regularizers we present the corresponding rates in Table 2.

With all this in hand, our main result can be stated as follows:

**Theorem 1.** *Let $x^*$ be a strict Nash equilibrium of $\Gamma$, and fix some confidence level $\delta > 0$. If Assumptions (A1)–(A3) hold, there exists an unbounded open set of initial conditions $\mathcal{W}_{\text{init}} \subseteq \mathcal{Y}$ and constants $d_i, c_i'$ with $c_i' > 0$ such that, if $Y_1 \in \mathcal{W}_{\text{init}}$, we have:*

*1. $X_n$ converges to $x^*$ with probability at least $1 - \delta$.*

*2. Conditioned on the above, the rate of convergence for each player $i \in \mathcal{N}$ is given by*

$$\|X_{i,n} - x_i^*\|_1 \le 2 \sum_{\alpha_i \in \mathcal{A}_i \setminus \text{supp}(x_i^*)} \phi_i\left(d_i - c_i' \sum_{k=1}^{n} \gamma_k\right). \tag{8}$$

Armed with this general result, we readily obtain the following immediate consequences thereof:

**Corollary 1.** *If the regularizer employed is non-steep (i.e., $\theta_i$ is differentiable at 0), $X_n$ converges to $x^*$ in a finite number of iterations.*

**Corollary 2.** *Suppose that FTRL is run with oracle-based feedback as per Model 1 and a learning rate of the form $\gamma_n \propto 1/n^p$, $p \in [0, 1]$. Then the conclusion of Theorem 1 holds.*

**Corollary 3.** *Suppose that FTRL is run with bandit feedback as per Model 2, a learning rate of the form $\gamma_n \propto 1/n^p$, $p \in [0, 1]$ and a mixing parameter $\varepsilon_n \propto 1/n^r$, $r \in (0, 1/2)$. Then the conclusion of Theorem 1 holds.*

*Remark* 4.1. We stress out here that for all the bandit-feedback derived results, the convergence rates refer to $X_{i,n}$ instead of the explicit exporation term $\hat{X}_{i,n}$ whose rate is always dominated by the mixing parameter $\varepsilon_n$.

**Corollary 4.** *Suppose that Optimistic FTRL is run with oracle-based feedback as per Model 3 and a learning rate of the form $\gamma_n \propto 1/n^p$, $p \in (0, 1]$. Then the conclusion of Theorem 1 holds.*

| ALGORITHM | KERNEL $\theta(\mathbf{x})$ | RATE $\phi(-\mathbf{y})$ |
|---|---|---|
| Multiplicative Weight Updates | $x \log x$ | $\exp(-y)$ |
| Projection Gradient Descent | $x^2/2$ | $-y$ |
| Inverse Updates | $-\log x$ | $-1/y$ |
| q-Replicator$_{q>0}$ | $[q(1-q)]^{-1}(x - x^q)$ | $[q^{-1} + (1-q)y]^{1/q-1}$ |

**Table 2:** Regularizers & corresponding rates.

More generally, we show in the supplement that the conclusion of Theorem 1 holds for all algorithms and feedback models presented in Table 1: in all cases therein, players can employ step-size policies of the form $\gamma_n \propto 1/n^p$, $p \in [0, 1]$, and a mixing parameter $\varepsilon_n \propto 1/n^r$ with $r \in (0, 1/2)$ for the bandit models. The only case that does not follow as an immediate corollary of Theorem 1 is the case of constant step-sizes for Optimistic FTRL and EG/MP; however, a slightly more refined argument (that we present in the **??**) shows that constant step-sizes are also covered by the convergence rate guarantee (8) of Theorem 1.

**4.2. Sketch of proof and main techniques.** At a high level, the main idea of the proof of Theorem 1 relies on a tandem application of martingale limit theory and convex analysis in order to exploit the specific structure of (FTGL). While martingale limit theory emerges naturally to control the components of the noise, a delicate analysis of the contribution of $h_i$ in the solution of the convex constrained optimization problem, that $x = Q_i(y)$ induces, is necessary to derive the aforementioned generic rates. Below we provide a sketch of the main steps in this analysis

**Step 1.** Our starting point is to explore the geometric properties that are induced by the existence of a strict Nash equilibrium. Indeed, the fact that (NE) holds as a strict inequality for each pure strategy against the equilibrium's strategy, ensures convergence properties for strict Nash equilibria. More precisely, an immediate implication of (NE) is that there exist neighborhood $\mathcal{U}$ of $x^*$ and constants $c_1, \ldots, c_N$ such that

$$v_{i\alpha_i^*}(x) - v_{i\alpha_i}(x) \geq c_i \text{ for all } x \in \mathcal{U} \text{ and } \alpha_i \neq \alpha_i^*, \alpha_i \in \mathcal{A}_i, i \in \mathcal{N} \tag{9}$$

In other words, in the neighborhood $\mathcal{U}$ the payoff of the equilibrium's strategy strictly dominates all other strategies' payoffs for each player. However, since the linchpin of (FTGL) is the interplay between $\mathcal{X}$ and $\mathcal{Y}$, for the purpose of our analysis, we need to investigate the variational structure of $\mathcal{U}$ in both spaces.

**Informal Lemma 1.** *There exists a neighborhood $\mathcal{U}$, constants $c_1, \ldots, c_N$ and $M_1, \ldots, M_N$ for which (9) holds such that $\prod_{i \in \mathcal{N}} Q_i(\mathcal{W}_{M_i}) \subseteq \mathcal{U}$, where $\mathcal{W}_{M_i}$ are open sets of the form* [3]

$$\mathcal{W}_{M_i} = \{Y_i : Y_{i\alpha_i^*} - Y_{i\alpha_i} > M_i \text{ for all } \alpha_i \neq \alpha_i^*, \alpha_i \in \mathcal{A}_i\} \text{ for } M_i > 0, i \in \mathcal{N} \tag{10}$$

**Step 2.** We now focus on one player $i \in \mathcal{N}$ and drop the index $i$ altogether. First we prove that there exists an open set of initializations $\mathcal{W}_{\text{init}}$ of (FTGL), for which with arbitrary high probability the dual variable $(Y_k)_{k \in \mathbb{N}}$ never exits $\mathcal{W}_M$ and thus its image remains in the desired neighborhood $\mathcal{U}$. We start by writing the score differences between each pure strategy $\alpha \in \mathcal{A}$ and $\alpha^* \in \text{supp}(x^*)$

$$Y_{\alpha,n+1} - Y_{\alpha^*,n+1} = Y_{\alpha,1} - Y_{\alpha^*,1} + \sum_{k=1}^{n} \gamma_k (\text{drift}_k + \text{noise}_k + \text{bias}_k) \tag{11}$$

where $\text{drift}_k = v_\alpha(X_k) - v_{\alpha^*}(X_k), \text{noise}_k = U_{\alpha,k} - U_{\alpha^*,k}, \text{bias}_k = b_{\alpha,k} - b_{\alpha^*,k}$. We will prove by induction our forward-invariant statement; let $Y_k \in \mathcal{W}_M$ and thus $X_k \in \mathcal{U}$ for all $k = 1, \ldots, n$ then

- By (9) we have $\sum_{k=1}^{n} \gamma_k \text{drift}_k \leq -c \sum_{k=1}^{n} \gamma_k$ for all $k = 1, \ldots, n$.
- By the triangle inequality and (A1), the term $\sum_{k=1}^{n} \gamma_k \text{bias}_k$ is dominated by the term $\sum_{k=1}^{n} \gamma_k \text{drift}_k$ for all $n = 1, 2, \ldots$.
- Subsequently, by leveraging the machinery of martingale's maximal inequalities and assumption (A2), which we present in **??** and using learning rates that respect (A3), we prove that with probability at least $1 - \delta$, for any fixed confidence level $\delta$, $\sum_{k=1}^{n} \gamma_k \text{noise}_k$, which is a martingale, is also dominated by the term $\sum_{k=1}^{n} \gamma_k \text{drift}_k$ for all $n = 1, 2, \ldots$
- We now define the open set of initial conditions $\mathcal{W}_{\text{init}}$, which is of the form described in (10), with constant $M_{\text{init}}$. By choosing[4] $M_{\text{init}} \geq M + \sum_{k=1}^{n} \gamma_k (\text{noise}_k + \text{bias}_k) - (c - c') \sum_{k=1}^{n} \gamma_k$, for any $c' < c$ and any $n \geq 1$, since $Y_1 \in \mathcal{W}_{\text{init}}$ we have that $Y_{\alpha,n+1} - Y_{\alpha^*,n+1} \leq -M$ for all $n \geq 1$.

---

[3] It is worth mentioning that the images of these open sets belong to neighborhoods of $x^*$, which are nested as $M_i$ increases.

[4] such a $M_{\text{init}}$ exists since both the bias and the noise terms are dominated by the term $-(c - c') \sum_{k=1}^{n} \gamma_k$.

By substituting the inequality for $M_{\text{init}}$ in (11) we get $Y_{\alpha,n+1} - Y_{\alpha^*,n+1} \leq -M - c' \sum_{k=1}^{n} \gamma_k$ and convergence occurs as an immediate consequence; Indeed $X_{\alpha^*,n} \to 1$, since whenever $Y_\alpha - Y_{\alpha^*} \to -\infty$, it holds that each $\alpha \in \mathcal{A} \setminus \text{supp}(x^*)$ becomes extinct i.e., $X_\alpha \to 0$.

**Step 3.** We now proceed to the delineation of the rates of convergence. Using the KKT conditions (**??**) combined with Eq. (11),Eq. (9) and the fact that $Y_1 \in \mathcal{W}_{\text{init}}$ we have

$$\theta'(X_{\alpha,n+1}) - \theta'(X_{\alpha^*,n+1}) = Y_{\alpha,n+1} - Y_{\alpha^*,n+1} \leq -M_{\text{init}} - c\sum_{k=1}^{n} \gamma_k + \sum_{k=1}^{n} \gamma_k(\text{noise}_k + \text{bias}_k)$$

Recall that $\theta$ is strong convex, or equivalently $\theta'$ is strictly increasing; by rearranging and substituting to the above inequality we get

$$\theta'(X_{\alpha,n+1}) \leq \theta'(X_{\alpha^*,n+1}) - M - c'\sum_{k=1}^{n} \gamma_k \leq d - c'\sum_{k=1}^{n} \gamma_k \tag{12}$$

where $d = -M + \theta'(1)$ and $\alpha \in \mathcal{A}, \alpha \neq \alpha^*$. By aggregating over all $\alpha \in \mathcal{A}, \alpha \neq \alpha^*$

$$\|x^* - X_{n+1}\|_1 = 2(1 - X_{\alpha^*,n+1}) \leq 2\sum_{\alpha \in \mathcal{A} \neq \alpha^*} \phi(d - c'\sum_{k=1}^{n} \gamma_k) \tag{13}$$

which indicates the rate of convergence and completes our proof.

*Remark* 4.2. *The bounds we provide are indeed sharp. To see this, consider a single-player game with two actions "0" and "1", and payoffs $u(0) = 0$, $u(1) = 1$. Then, if e.g., FTRL is run with "full information" feedback, the probability that the player plays "1" at time t is exactly equal to*

$$X_t = 1 - \phi(c - \sum_{s=1}^{t} \gamma_s u(1)) = 1 - \phi(c - \sum_{s=1}^{t} \gamma_s)$$

*where $\phi$ is the rate function of Eq. (7) and c is an initialization constant. This simple derivation shows that MWU converges to the game's (strict) equilibrium at a rate of exactly $\exp(-\Theta(\sum_{s=1}^{t} \gamma_s))$, whereas Euclidean methods achieve an equilibrium after a finite number of iterations – in particular, as soon as $\sum_{s=1}^{t} \gamma_s$ exceeds c. It thus follows that the rates provided by Theorem 1 are, in general, unimprovable.*

## 5 Numerical experiments

In this section we perform a series of numerical experiments to validate our theoretical findings. Specifically we are interested in verifying both the correctness in the computation of the rates via $\phi_i$ for different regularizers and at the same time the fact that convergence speed is invariant to different feedback models and algorithmic variants of (FTGL).

To do this, we start by examining variations of (FTGL) in the archetypal game of *Battle of the Sexes*, a popular two-player benchmark of the coordination games, which however involves elements of conflict as well. This game exhibits two strict Nash equilibria and one mixed equilibrium (for the exact definition, see **??**). We then seek to experimentally study the performance of (FTGL) while the number of the players scales up. To do this we use the atomic version of classical *Pigou's congestion game* [37], where $N$ units of traffic (e.g., rush-hour drivers) leave from $O$ (origin) to $D$ (destination) at the same time and each driver has the same dominant pure strategy/path for this trip. Accordingly to Table 2 the decay rate for the entropic regularization is exponential while for the case of euclidean is linear, which indeed yield linear and constant-time convergence as Fig. 1 illustrates.

We defer a detailed exposition of various configurations with different step-sizes, alternative discretization methods like MirrorProx and ExtraGradient and feedback models with the presence (or not) of extra heavy-tailed/uniform/gaussian noise again to the paper's supplement.

It is worth mentioning that the sharpness of the provided rates of Theorem 1 can clearly be observed in the list of the extensive numerical experiments we present in Fig. 1 and **??**. In particular, the faster convergence rate of Euclidean algorithms is somewhat surprising since a regret-based viewpoint would suggest the use of entropic regularization (which, ceteris paribus, has much better regret guarantees) as optimal in this regard. Interestingly, however our analysis shows that a Euclidean

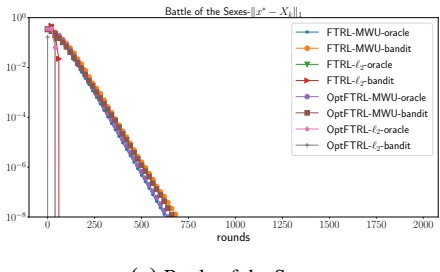
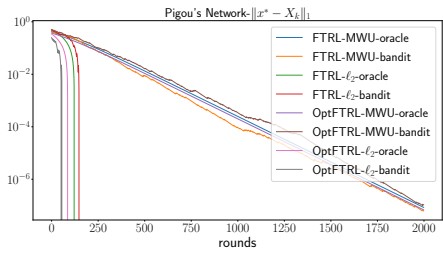

**(a)** Battle of the Sexes

**(b)** Pigou Network

**Figure 1:** For the *Battle of the Sexes* experiment, we initialize uniformly randomly our executions from $Y_{init} \in [-1, 1] \times [-1, 1]$ and examine the instantiations of Model 1-3 using constant-step size and exploration rate $\varepsilon_n \propto 1/\sqrt[3]{n}$. For the *Pigou's* game, our setup includes two alternative disjoint paths for $N = 1000$ drivers. The first path has linear latency $\ell_1(x) = x/N$ while the second one has constant unit congestion, $\ell_2(x) = 1$, where $x$ denotes the population of the drivers that uses the corresponding path.

regularizer is much more suitable for achieving convergence to equilibrium in a game-theoretic setting. It is for this reason that we insisted on the comparison between entropic and Euclidean regularization in the simulations. (*The Pigou network example of Fig. 1b is especially telling in this regard.*)

## 6    Concluding remarks

A key take-away from this study is that the questions of regret minimization and convergence to Nash equilibrium are fundamentally different. In particular, much of the conventional wisdom that has been accrued for regret-minimization strategies (such as which regularizer to use, with what learning rate, etc.) ceases to apply when the figure of merit is convergence to an equilibrium. Because the only states that are stable under leader-following policies are the game's strict Nash equilibria, the agents can be significantly more firm and confident in their choices, without compromising their final limit state; as a result, this extra degree of "confidence" allows for rates of convergence that are well beyond the operational envelope of regret minimization problems. We believe that this polar shift in perspective constitutes an important - and under-explored - issue in game-theoretic learning, and charting out its ramifications for multi-agent learning is a particularly fruitful direction for future research.

## Acknowledgments and Disclosure of Funding

This research was partially supported by the COST Action CA16228 "European Network for Game Theory" (GAMENET) and the Onassis Foundation under Scholarship ID: F ZN 010-1/2017-2018. P. Mertikopoulos is also grateful for financial support by the French National Research Agency (ANR) in the framework of the "Investissements d'avenir" program (ANR-15-IDEX-02), the LabEx PERSYVAL (ANR-11-LABX-0025-01), MIAI@Grenoble Alpes (ANR-19-P3IA-0003), and the grant ALIAS (ANR-19-CE48-0018-01). E.V. Vlatakis-Gkaragkounis is grateful to be supported by NSF grants CCF-1703925, CCF1763970, CCF-1814873, CCF-1563155, and by the Simons Collaboration on Algorithms and Geometry.

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
