# A  Martingale limit theory

Our analysis leverages tools from martingale limit theory. Below we present the two main theorems that we utilize in the main body of our proofs.
- (Doob's inequality), also known as *Kolmogorov's submartingale inequality* gives a bound on the probability that a stochastic process exceeds any given value over a given interval of time.
- (Burkholder's inequality), also known the *Burkholder-Davis-Gundy inequality* is a remarkable result relating the maximum of a local martingale with its quadratic variation.

**Theorem A.1** (Doob's inequality)**.** *Let $S_n$ be a martingale with respect to the filtration $\mathcal{F}_n$, then for each $\varepsilon > 0$ and $q \geq 1$,*

$$\mathbb{P}(\sup_{1 \leq k \leq n} |S_k| \geq \varepsilon) \leq \frac{\mathbb{E}|S_n|^q}{\varepsilon^q} \qquad \text{(Doob's inequality)}$$

**Theorem A.2** (Burkholder's inequality)**.** *Let $S_n$ be a martingale with respect to the filtration $\mathcal{F}_n$ and $X_n = S_n - S_{n-1}$. Then for all $1 < q < \infty$, there exists constant $C_q$ depending only on $q$ such that*

$$\mathbb{E}|S_n|^q \leq C_q \, \mathbb{E} \left| \sum_{k=1}^{n} X_k^2 \right|^{q/2} \qquad \text{(Burkholder's inequality)}$$

Proofs for these two theorems can be found in [19].

# B  A dichotomy between the regularizers

Our main result (Theorem 1) provides a mechanism to compute the convergence rate to a strict Nash Equilibrium universally for all smooth convex regularizers $h_i(x) = \sum_{\alpha_i \in \mathcal{A}_i} \theta_i(x_{\alpha_i})$. An important implication of our main theorem (Corollary 1) is that for the case of non-steep kernels (i.e., $\theta_i$ is differentiable at 0), $X_n$ converges to $x^*$ in a finite number of iterations. Below we give some intuition for the interested reader about the differences between the *steep* and *non-steep* case.

**Steep vs non-steep.**   In this section we elaborate in detail the dichotomy among the different regularizers mentioned in Sections 3.1 and 4. As we established in Section 3.1, different players may apply different regularizers $h_i$ in their choice maps $Q_i(y_i)$. Depending on the regularizer chosen, the behavior of (FTGL) could vary significantly. To investigate more this diversity, we start by describing formally the strategy-choice step $x_i = Q_i(y_i)$ as a convex constrainted minimization problem.

$$Q_i(y_i) = -\arg\min_{x_i \in \mathcal{X}_i} \{h_i(x_i) - \langle x_i, y_i \rangle\}. \tag{B.1}$$

Following also the folklore convention from convex analysis, we express $h$ as an extended-real valued function $h : \mathcal{V} \to \mathbb{R} \cup \{\infty\}$ with value $\infty$ outside of the simplex $\mathcal{X}$. Additionally, the subdifferential of $h$ at $x \in \mathcal{V}$ is defined as:

$$\partial h(x) = \{y \in \mathcal{V}^* : h(x') \geq h(x) + \langle y, x' - x \rangle \ \forall x' \in \mathcal{V}\} \tag{B.2}$$

If $\partial h(x)$ is nonempty, then $h$ is called subdifferentiable at $x \in \mathcal{X}$. When $x \in \mathrm{ri}(\mathcal{X})$ then $\partial h(x)$ is always non-empty or more compactly $\mathrm{ri}(\mathcal{X}) \subseteq \mathrm{dom}\,\partial h \equiv \{x \in \mathcal{X} : \partial h(x) \neq \emptyset\} \subseteq \mathrm{dom}\,h \subseteq \mathcal{X}$. Notice that when the gradient of $h$ exists, then its subgradient always contains it. Leveraging the property that local and global minima coincides in the case of convex function, Fermat's rule provides a simple characterization of the minimizers of a function as the zeros of its subdifferential:

**Fact** (Fermat's Rule)**.** *For a proper convex function $f$, $\arg\min f \equiv \mathrm{zer}\,\partial f = \{x \in \mathcal{X} \mid 0 \in \partial f(x)\}$*

With these in mind, we present a typical separation between the different regularizers,, focusing on the more simple case of decomposable ones $h(x) = \sum_{\alpha \in \mathcal{A}} \theta_\alpha(x)$. On the one hand, *steep* regularizers have differentiable kernels on $(0, 1]$ and become infinitely steep as $x$ approaches the boundary or $\theta'(0) = -\infty$. On the other hand, for the *non-steep* case the kernel is differentiable in all of $[0, 1]$. As a result of Fermat's Rule, when a steep regularizer is employed the points of the boundary are

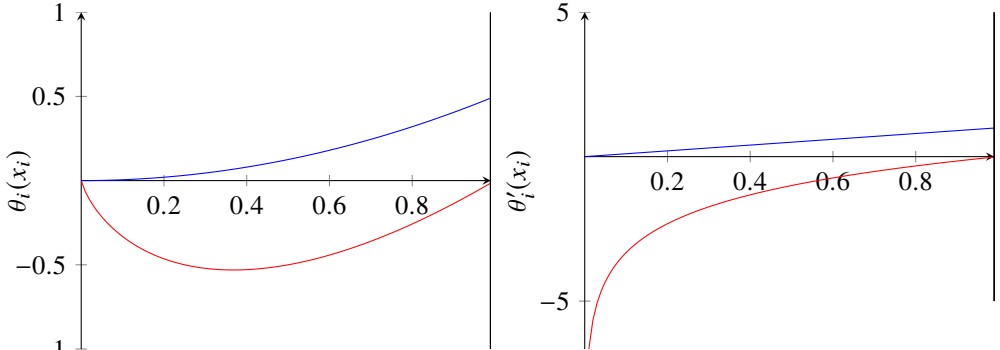

$$\begin{bmatrix} \text{steep} & h_1(x) & = & \displaystyle\sum_{\alpha \in \mathcal{A}} \theta_i(x_\alpha) & = & \displaystyle\sum_{\alpha \in \mathcal{A}} x_\alpha \log(x_\alpha) \\ \text{non-steep} & h_2(x) & = & \displaystyle\sum_{\alpha \in \mathcal{A}} \theta_i(x_\alpha) & = & \displaystyle\sum_{\alpha \in \mathcal{A}} \frac{1}{2} x_\alpha^2 \end{bmatrix}$$

**Figure 2:** Steep vs. non-steep regularizers (note in particular the singular behavior of the gradient at the boundary in the case of steep regularizers).

infeasible not only as initial conditions but also as part of the sequence of play, while non-steep ones allow completely the sequence of play to transfer between the different faces of the simplex. The qualititative difference in behavior between these cases is illustrated in Fig. 2 (which shows the very different behavior of the derivates of $h$ near the boundary of the state space).

> Having discussed the connection between the choice map and the properties of the regularizer, the following lemma quantifies the gulf between the steep and non-steep case and provides the relation between mixed strategies and score vectors and the mirror map (3) that defines the dynamics (FTGL). More precisely, we focus on the perspective of an arbitrary player, say $i$, and for ease of notation we write $Q$, $x$ and $y$ instead of $Q_i$, $x_i$ and $y_i$ respectively.

**Lemma B.1.** $x = Q(y)$ *if and only if there exist* $\mu \in \mathbb{R}$ *and* $\nu_\alpha \in \mathbb{R}_+$ *such that, for all* $\alpha \in \mathcal{A}$, *we have:* a) $y_\alpha = \frac{\partial h}{\partial x_\alpha} + \mu - \nu_\alpha$; *and* b) $x_\alpha \nu_\alpha = 0$ *In particular, if $h$ is steep, we have $\nu_\alpha = 0$ for all* $\alpha \in \mathcal{A}$.

*Proof.* Recall that

$$Q(y) = \arg\max_{x \in \mathcal{K}} \{\langle y|x \rangle - h(x)\}$$

$$= \arg\max \left\{ \sum_{\alpha \in \mathcal{A}} y_\alpha x_\alpha - h(x) : \sum_{\alpha \in \mathcal{A}} x_\alpha = 1 \text{ and } \forall \alpha \in \mathcal{A} : x_\alpha \geq 0 \right\}$$

The result follows by applying the Karash-Kuhn Tucker (KKT) conditions to this optimization problem and noting that, since the constraints are affine, the KKT conditions are sufficient for optimality. Our Langragian is

$$\mathcal{L}(x, \mu, \nu) = \left( \sum_{\alpha \in \mathcal{A}} y_\alpha x_\alpha - h(x) \right) - \mu \left( \sum_{\alpha \in \mathcal{A}} x_\alpha - 1 \right) + \sum_{\alpha \in \mathcal{A}} \nu_\alpha x_\alpha$$

where the set of constraints (i) of the statement of the lemma are the stationarity constraints, which in our case are $\nabla \mathcal{L}(x, \mu, \nu) = 0 \Leftrightarrow \nabla(\sum_{\alpha \in \mathcal{A}} y_\alpha x_\alpha - h(x)) = \mu \nabla(\sum_{\alpha \in \mathcal{A}} x_\alpha - 1) - \sum_{\alpha \in \mathcal{A}} \nu_\alpha \nabla x_\alpha$, while the set of constraints (ii) of the statement of the lemmas are the complementary slackness constraints. Note that complementary slackness implies that whenever $\nu_\alpha > 0$ whenever $\alpha \notin \text{supp}(x)$. Finally, if $h$ is steep, we have $|\partial_\alpha h(x)| \to \infty$ as $x \to \text{bd}(\mathcal{X})$, which implies that the KKT conditions admit a solution with $\nu_\alpha = 0$. ∎

# C Proof of Main Theorem

Our first lemma shows a property of strict Nash equilibria. More precisely, we prove the existence of a neighborhood $\mathcal{U}$ in which each player's payoff corresponding to the strategy of the equilibrium outweighs the payoff of any other pure strategy.

**Lemma C.1.** *Let $x^* = (\alpha_1^*, \ldots, \alpha_N^*) \in \mathcal{A}$ be a strict Nash equilibrium. Then there exists a neighborhood $\mathcal{U}$ of $x^*$ and constants $c_i$ such that for each player $i \in \mathcal{N}$:*

$$v_{i\alpha_i^*}(x) - v_{i\alpha_i}(x) \geq c_i \text{ for all } x \in \mathcal{U} \text{ and } \alpha_i \neq \alpha_i^*, \alpha_i \in \mathcal{A}_i. \tag{C.1}$$

*Proof.* Our claim is a consequence of the definition of strict Nash equilibria. Specifically, from (NE) for each player $i \in \mathcal{N}$ we have that

$$v_{i\alpha_i^*}(x^*) > v_{i\alpha_i}(x^*) \text{ for all } \alpha_i \in \mathcal{A}_i, \alpha_i \neq \alpha_i^* \tag{C.2}$$

By continuity there exists a neighborhood $\mathcal{U} \subseteq \mathcal{X}$ and $c_i > 0$ for each player $i \in \mathcal{N}$ such that

$$v_{i\alpha_i^*}(x) - v_{i\alpha_i}(x) \geq c_i \text{ for all } x \in \mathcal{U} \tag{C.3}$$

■

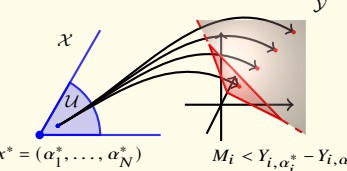

The following lemma plays a central role in the proof of our main theorem (Theorem 1). More precisely, Lemma C.2 provides a detailed analysis of the topology of a neighborhood $\mathcal{U}$ where variational inequality (C.1) holds both in primal space $\mathcal{X}$ and dual space $\mathcal{Y}$. In order to achieve that we introduce the notion of "$(\alpha_i^*, M_i)$-score-dominant" open set for a player $i \in \mathcal{N}$, which we denote $\mathcal{W}_i(M_i)$.

**Definition** (*Score-Dominant Collection*). Let $x^* = (\alpha_1^*, \ldots, \alpha_N^*) \in \mathcal{A}$ be a strict Nash equilibrium of a finite game $\Gamma$. Then a collection is said to be "$(\alpha_i^*, M_i)_{i \in \mathcal{N}}$-score-dominant" if there exist positive constants $M_i > 0$ corresponding open sets $\mathcal{W}_i(M_i)$ of the form

$$\mathcal{W}_i(M_i) = \{Y_i : Y_{i\alpha_i^*} - Y_{i\alpha_i} > M_i \text{ for all } \alpha_i \neq \alpha_i^*, \alpha_i \in \mathcal{A}_i\} \text{ for each player } i \in \mathcal{N} \tag{C.4}$$

**Lemma C.2.** *Let $x^* = (\alpha_1^*, \ldots, \alpha_N^*) \in \mathcal{A}$ be a strict Nash equilibrium. Then for every $\varepsilon \in (0, 1)$, there exist constants $M_{i,\varepsilon}$ and the corresponding score-dominant open sets for each player $i \in \mathcal{N}$ such that: $\prod_{i \in \mathcal{N}} Q_i(\mathcal{W}_i(M_{i,\varepsilon})) \subseteq \mathcal{U}_\varepsilon$, where $\mathcal{U}_\varepsilon = \{x \in \mathcal{X} : x_{i\alpha_i^*} > 1 - \varepsilon \text{ for every player } i \in \mathcal{N}\}$*

*Proof.* For an arbitrary player $i \in \mathcal{N}$ let $\mathcal{W}_i(M_{i,\varepsilon})$ be a *score-dominant* open set. We will show that any $M_{i,\varepsilon} > \theta_i'(1) - \theta_i'(\frac{\varepsilon}{|\mathcal{A}_i|}) > 0$ satisfies the desideratum. Indeed, again by using Lemma B.1 for a $Y_i \in \mathcal{W}_i(M_{i,\varepsilon})$ with $x_i = Q_i(Y_i)$ we have that

$$Y_{i\alpha_i^*} - Y_{i\alpha_i} > M_{i,\varepsilon} \tag{C.5}$$

$$\theta'(x_{i\alpha_i^*}) - \theta_i'(x_{i\alpha_i}) - (v_{\alpha_i^*} - v_{\alpha_i}) > M_{i,\varepsilon}. \tag{C.6}$$

with $v_{\alpha_i} \geq 0$ and $x_{i\alpha_i} = 0$ whenever $x_{i\alpha_i} > 0$. Notice that since $M_{i,\varepsilon} > 0$ and $\theta_i'$ is strictly increasing, it holds that $x_{i\alpha_i} < x_{i\alpha_i^*}$. Indeed, assume by contradiction that $x_{i\alpha_i} \geq x_{i\alpha_i^*}$ for some $\alpha_i$, then we examine two different cases:

(i) If $x_{i\alpha_i^*} = 0$, then $x_{i\alpha_i} \geq x_{i\alpha_i^*}$ for all $\alpha_i \in \mathcal{A}_i$ with $x_{i\alpha_i} > 0$ for at least one $\alpha_i \in \mathcal{A}_i, \alpha_i \neq \alpha_i^*$ which is a contradiction to (C.6).

(ii) if $x_{i\alpha_i^*} > 0$, then (C.6) implies that $M_{i,\varepsilon} \leq \theta'(x_{i\alpha_i^*}) - \theta_i'(x_{i\alpha_i}) < 0$ which is again a contradiction.

Therefore $v_{\alpha_i^*} = 0$ and (C.6) can be rewritten for all $\alpha_i \neq \alpha_i^*$ with $x_{i\alpha_i} > 0$ as

$$\theta_i'(x_{i\alpha_i}) < -M_{i,\varepsilon} + \theta'(x_{i\alpha_i^*}) < -M_{i,\varepsilon} + \theta'(1) < \theta_i'(\frac{\varepsilon}{|\mathcal{A}_i|}) \tag{C.7}$$

where last inequality holds by the choice of $M_{i,\varepsilon} > \theta_i'(1) - \theta_i'(\frac{\varepsilon}{|\mathcal{A}_i|}) > 0$. Again, since $\theta'$ is strictly increasing, this implies that for all $\alpha_i \neq \alpha_i^*$ either $x_{i\alpha_i} = 0$ or $0 < x_{i\alpha_i} < \frac{\varepsilon}{|\mathcal{A}_i|}$. By union bound, this implies that $x_{i\alpha_i^*} > 1 - \varepsilon$ and equivalently that $x \in \mathcal{U}_\varepsilon$.

■

*Remark* C.1. It is easy to check that as $M_i'$ increases the score-dominant sets and their corresponding images are nested. Indeed if $M' \geq M_\varepsilon \Rightarrow \mathcal{W}(M) \subseteq \mathcal{W}(M') \Rightarrow Q(\mathcal{W}(M)) \subseteq Q(\mathcal{W}(M'))$, since $Y_{i\alpha_i^*} - Y_{i\alpha_i} > M > M_\varepsilon$ for all $\alpha_i \neq \alpha_i^*, \alpha_i \in \mathcal{A}_i$.

*Remark* C.2. Notice that since the above analysis is for each strategy $\alpha_i \in \mathcal{A}_i$ of player $i$, it implies that not only the images $Q_i(\mathcal{W}_{M_i})$ are nested, but also that if $x_i = Q_i(Y_i)$, $Y_i \in \mathcal{W}_{M_i}$ all $x_{i\alpha_i} \to 0$ for $\alpha_i \neq \alpha_i^*$ as $M_i \to \infty$.

**Theorem 1.** *Let $x^*$ be a strict Nash equilibrium of $\Gamma$, and fix some confidence level $\delta > 0$. If Assumptions (A1)–(A3) hold, there exists an unbounded open set of initial conditions $\mathcal{W}_{\mathrm{init}} \subseteq \mathcal{Y}$ and constants $d_i, c_i'$ with $c_i' > 0$ such that, if $Y_1 \in \mathcal{W}_{\mathrm{init}}$, we have:*

1. *$X_n$ converges to $x^*$ with probability at least $1 - \delta$.*
2. *Conditioned on the above, the rate of convergence for each player $i \in \mathcal{N}$ is given by*

$$\|X_{i,n} - x_i^*\|_1 \leq 2 \sum_{\alpha_i \in \mathcal{A}_i \setminus \mathrm{supp}(x_i^*)} \phi_i\left(d_i - c_i' \sum_{k=1}^n \gamma_k\right). \tag{8}$$

*Remark* C.3. The probability guarantee is over only the potential randomness that the payoff oracle. i.e., when players have access to a perfect payoff oracle; the results hold with probability 1.

*Proof.* Fix a confidence level $\delta$ and the parameters of the algorithm respecting (A1)–(A3). We will prove that there exists a "score-dominant" open set of initial conditions $\mathcal{W}_{\mathrm{init}}$

$$\mathcal{W}_{\mathrm{init}} \equiv \{Y : M_{\mathrm{init}} < Y_{\alpha^*} - Y_\alpha \text{ for all } \alpha \neq \alpha^*, \alpha \in \mathcal{A}\} \subseteq \mathcal{Y} \text{ for some } M_{\mathrm{init}} > 0$$

such that whenever $Y_1 \in \mathcal{W}_{\mathrm{init}}$ then with probability at least $1 - \delta$ the sequence of play generated by (FTGL) converges to $x^*$ with rate given by the function $\phi_i$

$$\phi_i(t) = \begin{cases} (\theta_i')^{-1}(t) & \text{if } t > \theta_i'(0^+), \\ 0 & \text{otherwise.} \end{cases} \tag{C.8}$$

which depends on the choice of the kernel $\theta_i$ of each player and the payoff matrix of the game.

For convenience of notation we focus on an arbitrary player in the proof, without loss of generality let it be the $i$-th one, and we completely drop the index $i$. Since the equilibrium is strict by Lemmas C.1 and C.2 there exist a neighborhood $\mathcal{U}_{\mathrm{strict}}$, $c_{\mathrm{strict}} > 0$ and $M_{\mathrm{strict}} > 0$ such that

$$v_{\alpha^*}(x) - v_\alpha(x) \geq c_{\mathrm{strict}} \quad \text{for all } \alpha \neq \alpha^*, \alpha \in \mathcal{A} \text{ and } x \in \mathcal{U}_{\mathrm{strict}} \tag{C.9}$$

$$Y_\alpha^* - Y_\alpha > M_{\mathrm{strict}} \quad \text{for all } \alpha \neq \alpha^*, \alpha \in \mathcal{A} \text{ and } x = Q(Y) \in \mathcal{U}_{\mathrm{strict}} \tag{C.10}$$

We start by proving the following claim:

**Claim 1.** *Let $\mathcal{W}(M)$ be a "score-dominant" open set for the strict Nash equilibrium $x^*$. Then there exists $M_{\mathrm{init}} > 0$ such that if $Y_1 \in \mathcal{W}(M_{\mathrm{init}}) = \mathcal{W}_{\mathrm{init}}$ then with probability at least $1 - \delta$ the sequence of play $(Y_n)_{n \in \mathbb{N}}$ stays in $\mathcal{W}(M_{\mathrm{strict}})$.*

*Proof of Claim.* By definition of (FTGL) for the score differences we have

$$Y_{\alpha,n+1} - Y_{\alpha^*,n+1} = Y_{\alpha,1} - Y_{\alpha^*,1} + \sum_{k=1}^n \gamma_k (\mathrm{drift}_k + \mathrm{noise}_k + \mathrm{bias}_k) \tag{C.11}$$

where $\mathrm{drift}_k = v_\alpha(X_k) - v_{\alpha^*}(X_k)$, $\mathrm{noise}_k = U_{\alpha,k} - U_{\alpha^*,k}$, $\mathrm{bias}_k = b_{\alpha,k} - b_{\alpha^*,k}$. Notice that

- *(Bias)* By (A1): $\sum_{k=1}^n \gamma_k \mathrm{bias}_k \leq 2 \sum_{k=1}^n \gamma_k \|b_k\|_* = o(\sum_{k=1}^n \gamma_k)$ \hfill (C.12)

- *(Payoff)* By Lemma C.1: $\sum_{k=1}^n \gamma_k \mathrm{drift}_k \leq -c \sum_{k=1}^n \gamma_k$ \hfill (C.13)

- *(Zero-mean Noise)* For the remaining term, $R_n = \sum_{k=1}^n \gamma_k \mathrm{noise}_k$, firstly notice that it is trivially a martingale. We will prove that with probability at least $1 - \delta$ this martingale is bounded above by a term $\xi_n$ which is dominated by the term $\sum_{k=1}^n \gamma_k$. Consider the event $D_{n,\xi_n} = \{\sup_{1 \leq k \leq n} |R_k| \geq \xi_n\}$; we will show that the union of these events $\mathcal{E} = \bigcup_{n=1}^\infty D_{n,\xi_n}$ occurs with probability at most $\delta$ when $\xi_n = \xi(\sum_{k=1}^n \gamma_k)^a$ with $a < 1$. Using Theorem A.1 and Theorem A.2 we have

$$\mathbb{P}(D_{n,\xi_n}) \leq \frac{\mathbb{E}[|R_n|^q]}{\xi_n{}^q} \leq \frac{c_q \mathbb{E}[(\sum_{k=1}^n \gamma_k^2 \|U_k\|_*^2)^{q/2}]}{\xi_n^q} \tag{C.14}$$

**Fact** (Generalized Hölder's Inequality). *We will now consider a variation of the Hölder's inequality*

$$\left(\sum_{k=1}^{n} a_k b_k\right)^r \le \left(\sum_{k=1}^{n} a_k^{\frac{\mu r}{r-1}}\right)^{r-1} \sum_{k=1}^{n} a_k^{(1-\mu)r} b_k^r \text{ for all } r > 1, \mu \in (0,1) \tag{GH}$$

Applying (GH) for $a_k = \gamma_k^2$, $b_k = \|U_k\|_*^2$, $r = q/2$ and $\mu = (r-1)/2r = (q-2)/2q$, we get

$$\mathbb{P}(D_{n,\xi_n}) \le \frac{c_q (\sum_{k=1}^{n} \gamma_k)^{\frac{q-2}{2}} \sum_{k=1}^{n} \gamma_k^{1+q/2} \mathbb{E}[\|U_k\|_*^q]}{\xi_n^q} \tag{C.15}$$

$$\le \frac{c_q (\sum_{k=1}^{n} \gamma_k)^{\frac{q-2}{2}} \sum_{k=1}^{n} \gamma_k^{1+q/2} \mathbb{E}[\mathbb{E}[\|U_k\|_*^q \mid \mathcal{F}_k]]}{\xi_n^q} \tag{C.16}$$

$$\le \frac{c_q (\sum_{k=1}^{n} \gamma_k)^{\frac{q-2}{2}} \sum_{k=1}^{n} \gamma_k^{1+q/2} \sigma_k^q}{\xi_n^q} \tag{C.17}$$

Recall that $\xi_n = \xi \left(\sum_{k=1}^{n} \gamma_k\right)^a$ with $a < 1$ and let us denote $\delta_n = \frac{c_q}{\xi^q} \frac{\sum_{k=1}^{n} \gamma_k^{1+\frac{q}{2}} \sigma_k^q}{\left[\sum_{k=1}^{n} \gamma_k\right]^{1+(2a-1)q/2}}$ or

equivalently $\delta_n = \frac{c_q}{\xi^q} \frac{\sum_{k=1}^{n} \gamma_k^{1+\frac{q}{2}} \sigma_k^q}{\left[\sum_{k=1}^{n} \gamma_k\right]^{1+\beta q/2}}$ for some $\beta < 1$. By assumption (A3), $\delta_n$ is summable and by controlling the parameter $\xi$ we can ensure that

$$\sum_{n=1}^{\infty} \delta_n = \delta \tag{C.18}$$

Applying union bound to all the events $D_{n,\xi_n}$ we have that with probability at least $1 - \delta$ it is

$$\sum_{k=1}^{n} \gamma_k \text{noise}_k \le \xi_n \text{ for all } n = 1, 2, \dots \tag{C.19}$$

For the rest of the proof we condition to the event $\mathcal{E}^c$. Let us define a constant $M_{\text{init}}$, such that $M_{\text{init}} \ge \max\{M_{\text{strict}}, M_{\text{strict}} + \sup_{n \ge 1}\{\sum_{k=1}^{n} \gamma_k(\text{noise}_k + \text{bias}_k) - (c - c')\sum_{k=1}^{n} \gamma_k\}$, for any arbitray choice of $0 < c' < c_{\text{strict}}$ [5]. Let us recall the definition of a "score-dominant" open set

$$\mathcal{W}(M) = \{Y : Y_\alpha^* - Y_\alpha > M \text{ for all } \alpha \ne \alpha^*, \alpha \in \mathcal{A}\}.$$

We will prove by strong induction that $Y_n \in \mathcal{W}(M_{\text{strict}})$, for all $n \ge 1$.

- For the base of the induction, we have that $Y_1 \in \mathcal{W}(M_{\text{init}})$ and by the choice of $M_{\text{strict}}$, trivially we get that $Y_1 \in \mathcal{W}(M_{\text{strict}})$.
- For the inductive step, let us assume that $Y_k \in \mathcal{W}(M_{\text{strict}})$ for all $k = 1, 2, \dots, n$, we will show below that $Y_{n+1} \in \mathcal{W}(M_{\text{strict}})$.

Combining (C.12),(C.13),(C.19) for the terms $\sum_{k=1}^{n} \gamma_k \text{drift}_k$, $\sum_{k=1}^{n} \gamma_k \text{noise}_k$, $\sum_{k=1}^{n} \gamma_k \text{bias}_k$ the claim's assumption $Y_1 \in \mathcal{W}(M_{\text{strict}})$ and the choice of $M_{\text{init}}$, (C.11) can be bounded as

$$Y_{\alpha,n+1} - Y_{\alpha^*,n+1} = Y_{\alpha,1} - Y_{\alpha^*,1} + \sum_{k=1}^{n} \gamma_k(\text{drift}_k + \text{noise}_k + \text{bias}_k) \tag{C.20}$$

$$Y_{\alpha,n+1} - Y_{\alpha^*,n+1} \le Y_{\alpha,1} - Y_{\alpha^*,1} - c_{\text{strict}} \sum_{k=1}^{n} \gamma_k + \xi_n + 2\sum_{k=1}^{n} \gamma_k \|b_k\|_* \tag{C.21}$$

$$Y_{\alpha,n+1} - Y_{\alpha^*,n+1} \le -M_{\text{init}} - (c_{\text{strict}} - c')\sum_{k=1}^{n} \gamma_k + \xi_n + 2\sum_{k=1}^{n} \gamma_k \|b_k\|_* - c'\sum_{k=1}^{n} \gamma_k \tag{C.22}$$

$$Y_{\alpha,n+1} - Y_{\alpha^*,n+1} \le -M_{\text{strict}} - c'\sum_{k=1}^{n} \gamma_k \le -M_{\text{strict}} \tag{C.23}$$

and thus $Y_{n+1} \in \mathcal{W}(M_{\text{strict}})$. ∎

---

[5]such a $M_{\text{init}} > 0$ exists since both the bias and the noise terms are dominated by the term the terms $2\sum_{k=1}^{n} \gamma_k \|b_k\|_*, \xi_n$ and consequently by $-(c - c')\sum_{k=1}^{n} \gamma_k$.

The above claim immediately implies that $X_n \in \mathcal{U}$ for all $n = 1, 2, \ldots$. We will now prove that the sequence of play converges to $x^*$.

*Proof of Convergence.* Let's assume that ad absordum that there exists at least one strategy $\alpha \neq \alpha^*, \alpha \in \mathcal{A}$ such that $\limsup_{n \to \infty} X_{\alpha,n} \geq \varepsilon > 0$. for all sufficiently large $n$. Recall also that for $X \in \mathcal{U}_{\text{strict}}$, it holds that $X_{\alpha^*} > 0$ by construction in Lemma C.2.

Then by Lemma B.1 we have

$$Y_\alpha = \theta'(X_\alpha) + \mu - v_\alpha \tag{C.24}$$

where $\mu \in \mathbb{R}$ and $v_\alpha \geq 0$ while $v_\alpha = 0$ whenever $X_\alpha > 0$. Leveraging that *i)* the sequence of play is contained in $\mathcal{U}$, *ii)* by descending to a subsequence if necessary $X_{\alpha,m_i} > 0$ and *iii)* recall (C.23) for the subsequence we have

$$Y_{\alpha,m_{i+1}} - Y_{\alpha^*,m_{i+1}} = \theta'(X_{\alpha,m_{i+1}}) - \theta'(X_{\alpha^*,m_{i+1}}) \leq -M_{\text{strict}} - c' \sum_{k=1}^{m_i} \gamma_k \tag{C.25}$$

However, the RHS of the above inequality goes to $-\infty$ as $m_i \to \infty$, while the LHS of the above inequality is bounded by the constant $\theta'(\varepsilon) - \theta'(1)$ since $\theta'$ is strictly increasing, which is a contradiction[6]. ∎

*Proof of Rate.* We now proceed to the delineation of the exact rates achieved. Consider the function

$$\phi(t) = \begin{cases} (\theta')^{-1}(t) & \text{if } t > \theta'(0^+), \\ 0 & \text{otherwise.} \end{cases} \tag{C.26}$$

where $(\theta')^{-1}(z)$ is the inverse function of the kernel $\theta'$[7]. Focusing on (C.25) we can derive that

$$\theta'(X_{\alpha,n+1}) \leq -M_{\text{strict}} + \theta'(X_{\alpha^*,n+1}) - c' \sum_{k=1}^{n} \gamma_k \tag{C.27}$$

$$\leq -M_{\text{strict}} + \theta'(1) - c' \sum_{k=1}^{n} \gamma_k \tag{C.28}$$

for all $\alpha \in \mathcal{A}_i$ and $n = 1, 2, \ldots$. As a result

$$X_{\alpha,n+1} \leq \phi\left(-M_{\text{strict}} + \theta'(1) - c' \sum_{k=1}^{n} \gamma_k\right) \tag{C.29}$$

Aggregating over all strategies $\alpha \in \mathcal{A}$, $\alpha \neq \alpha^*$ we have

$$\|x^* - X_{n+1}\|_1 = 2(1 - X_{\alpha^*,n+1}) \tag{C.30}$$

$$\leq \sum_{\alpha \in \mathcal{A} \neq \alpha^*} \phi\left(-M_{\text{strict}} + \theta'(1) - c' \sum_{k=1}^{n} \gamma_k\right) \tag{C.31}$$

$$\leq \sum_{\alpha \in \mathcal{A} \neq \alpha^*} \phi\left(d - c' \sum_{k=1}^{n} \gamma_k\right) \tag{C.32}$$

where $d = -M_{\text{strict}} + \theta'(1)$. ∎

∎

---

[6]The aforementioned by contradiction argument also provides a short proof of Remark C.2.

[7]$\theta'$ is strictly increasing and so does its inverse.

**Corollary 1.** *If the regularizer employed is non-steep (i.e., $\theta_i$ is differentiable at 0), $X_n$ converges to $x^*$ in a finite number of iterations.*

*Proof.* Additionally, in the case of non-steep regularizers we can prove that convergence occurs in finite time. More precisely, focusing on (C.28) and bearing in mind that $X_{\alpha,n+1} \geq 0$ for all $n = 1, 2, \ldots$ we have

$$\theta'(0) \leq \theta'(X_{\alpha,n+1}) \leq -M_{\text{strict}} + \theta'(1) - c' \sum_{k=1}^{n} \gamma_k \tag{C.33}$$

At the same time for finite $n$ it holds

$$\sum_{k=1}^{n} \gamma_k \geq (-M_{\text{strict}} + \theta'(1) - \theta'(0))/c' \tag{C.34}$$

since $\theta'(0)$ is finite for non-steep regularizers. Rearranging the above inequality we have

$$-M_{\text{strict}} + \theta'(1) - c' \sum_{k=1}^{n} \gamma_k \leq \theta'(0) \tag{C.35}$$

which inevitably implies that $X_{\alpha,n+1} = 0$. ∎

# D   Models

We start by presenting the well-known algorithms *Follow the Regularized Leader* (FTRL), *Optimistic Follow the Regularized Leader* (OptFTRL) and *Mirror Prox* (MP), as special cases of our general algorithmic framework.

$$\begin{aligned} Y_{i,n+1} &= Y_{i,n} + \gamma_n V_{i,n} \\ X_{i,n} &= Q_i(Y_{i,n}) \end{aligned} \tag{FTRL}$$

$$\tilde{Y}_{i,n} = Y_{i,n} + \gamma_n V_{i,n-1} \qquad \tilde{X}_{i,n} = Q_i(\tilde{Y}_{i,n}) \qquad Y_{i,n+1} = Y_{i,n} + \gamma_n V_{i,n} \tag{OptFTRL}$$

*Remark* D.1. (OptFTRL) requires two initializations and then at each stage the previous payoff signal is stored and is utilized to calculate the auxiliary cumulative payoff $\tilde{Y}_{i,n}$.

$$\begin{aligned} Y_{i,n+1/2} &= Y_{i,n} + \gamma_n V_{i,n} & Y_{i,n+1} &= Y_{i,n} + \gamma_n V_{i,n+1/2} \\ X_{i,n+1/2} &= Q_i(Y_{i,n+1/2}) & X_{i,n+1} &= Q_i(Y_{i,n+1}) \end{aligned} \tag{MirrorProx}$$

*Remark* D.2. (MirrorProx) requires only one initialization, but at each stage the algorithm generates two different states and correspondingly two payoff signals are needed.

For both the algorithms (OptFTRL),(MirrorProx) we can prove that for the cases of full information, oracle based feedback and noisy payoff feedback, the implicit bias for modeling their intermediate steps is $\|b_{i,n}\|_* = \mathcal{O}(\gamma_n)$. The bias is the same in all of the three cases and thus we only present the case of full information.

*Proof.* **Full information:**

- (OptFTRL): $V_{i,n} = v_i(X_n) + (v_i(\tilde{X}_n) - v_i(X_n))$. Thus

$$\|b_{i,n}\|_* = \|v_i(\tilde{X}_n) - v_i(X_n)\|_* \leq C\|\tilde{X}_n - X_n\| \tag{D.1}$$

$$= C\|Q_i(\tilde{Y}_n) - Q_i(Y_n)\| \leq C'\|\tilde{Y}_n - Y_n\|_* \tag{D.2}$$

$$= \mathcal{O}(\gamma_n) \tag{D.3}$$

- (MirrorProx): $V_{i,n} = v_i(X_n) + (v_i(X_{n+1/2}) - v_i(X_n))$. The proof is similar to the above and $\|b_{i,n}\|_* = \mathcal{O}(\gamma_n)$.

∎

Below, we explain how the proof of Theorem 1 can be oriented to the specific structure of both (OptFTRL) and (MirrorProx), in order to achieve all the permitted step-sizes. We will not make an exact proof but we will thoroughly describe how the proof of Theorem 1 should be altered for the case of full information; the reader can follow similar steps for the case of oracle based feedback.

- *Optimistic Follow the Regularized Leader*
  (OptFTRL) has an extra auxiliary cumulative payoff $\tilde{Y}_n$. We will first prove that if the two initializations of (OptFTRL) are appropriate then Theorem 1 holds without introducing any bias term.
  **Step 1:** Notice that for the score differences of the auxiliary cumulative payoffs we have

$$\tilde{Y}_{\alpha,n+1} - \tilde{Y}_{\alpha^*,n+1} = Y_{\alpha,n} - Y_{\alpha^*,n} + \gamma_n\big(v_\alpha(\tilde{X}_{n-1}) - v_{\alpha^*}(\tilde{X}_{n-1})\big) \tag{D.4}$$

  By substituting all the $Y_n$ terms we have

$$\tilde{Y}_{\alpha,n+1} - \tilde{Y}_{\alpha^*,n+1} = Y_{\alpha,1} - Y_{\alpha^*,1} + \sum_{k=1}^{n-1} \gamma_k\big(v_\alpha(\tilde{X}_k) - v_{\alpha^*}(\tilde{X}_k)\big) + \gamma_n\big(v_\alpha(\tilde{X}_{n-1}) - v_{\alpha^*}(\tilde{X}_{n-1})\big) \tag{D.5}$$

  **Step 2:** Assume that $\tilde{Y}_k \in \mathcal{W}_M$ as described in Theorem 1 and thus $\tilde{X}_k \in \mathcal{U}$ for all $k = 1, \ldots, n$. We will prove by induction that $\tilde{Y}_{n+1} \in \mathcal{W}_M$. Notice that since $\tilde{X}_k \in \mathcal{U}$ it holds that

$$v_\alpha(\tilde{X}_k) - v_{\alpha^*}(\tilde{X}_k) \le -c \text{ for all } k = 1, \ldots, n \tag{D.6}$$

  **Step 3:** From Eq. (D.5) we have

$$\tilde{Y}_{\alpha,n+1} - \tilde{Y}_{\alpha^*,n+1} \le Y_{\alpha,1} - Y_{\alpha^*,1} - c\sum_{k=1}^{n} \gamma_k \tag{D.7}$$

  By choosing $M_{\text{init}} > M$ our claim follows. We stress here that we have implicitly assumed that for the second initialization of (OptFTRL) it holds $\tilde{Y}_1 \in \mathcal{W}$.
  **Step 4:** The rest of the proof holds as the one in Theorem 1, as all of the states $\tilde{X}_n$ remain in the desired neighborhood $\mathcal{U}$ in which the variational inequality holds.

- *Mirror Prox*
  This algorithm, as we have already mentioned, calculates two different cumulative payoffs and primal states at each round.
  **Step 1:** We will first prove by induction that that the cumulatve payoffs $Y_{n+1/2} \in \mathcal{W}_M$ for all $n = 1, 2, \ldots$. Assume that $Y_{k+1/2} \in \mathcal{W}_M$ and thus $X_{k+1/2} \in \mathcal{U}$ for all $k = 1, \ldots, n$ then for the score differences we have

$$Y_{\alpha,n+1/2} - Y_{\alpha^*,n+1/2} = Y_{\alpha,n} - Y_{\alpha^*,n} + \gamma_n(v_\alpha(X_n) - v_{\alpha^*}(X_n)) \tag{D.8}$$

$$= Y_{\alpha,1} - Y_{\alpha^*,1} + \sum_{k=1}^{n-1} \gamma_k\big(v_\alpha(X_{k-1/2}) - v_{\alpha^*}(X_{k-1/2})\big) \tag{D.9}$$

$$+ \gamma_n(v_\alpha(X_n) - v_{\alpha^*}(X_n)) \tag{D.10}$$

$$\le Y_{\alpha,1} - Y_{\alpha^*,1} - c\sum_{k=1}^{n-1} \gamma_k + \gamma_n \max_{\alpha \in \mathcal{A}} \|v(\alpha)\|_* \tag{D.11}$$

  **Step 2:** Choose $M_{\text{init}} > M + \gamma_n \max_{\alpha \in \mathcal{A}}\{\|v(\alpha)\|_*\}$ which is feasible for step-size of the form $\gamma_n \propto 1/n^p$, $p \in [0, 1]$ and our claim follows.
  **Step 3:** Continue with the proof as presented in Theorem 1.

Below we prove some properties concerning the case of payoff oracle/bandit feedback.

**Proposition D.1.** *In the bandit case, let $\tilde{X}_n$ be the state such that $\hat{X}_{i,n}$ is the mixed strategy of the $i^{th}$ player at round n i.e., $\hat{X}_{i,n} = (1 - \varepsilon_n)\tilde{X}_{i,n} + \varepsilon_n/|\mathcal{A}_i|$, based on which the pure strategy $\alpha_{i,n}$ is selected. Then the following properties hold*

1. $\mathbb{E}[U_{i,n} \mid \mathcal{F}_n] = 0$.
2. $\|U_{i,n}\|_* = \mathcal{O}(1/\varepsilon_n)$.
3. $\|b_{i,n}\|_* = \mathcal{O}(\varepsilon_n)$.

*Remark* D.3. In the case of (MirrorProx) $\tilde{X}_{i,n}$ is the state $X_{i,n-1/2}$.

*Proof.* The payoff signal which is estimated through the (IWE) can be rewritten as $V_{i,n} = v_i(X_n) + U_{i,n} + b_{i,n}$, where $U_{i,n} = V_{i,n} - v_i(\hat{X}_n)$ and $b_{i,n} = v_i(\hat{X}_n) - v_i(X_n)$.

1. Let $\mathcal{A}_i = \{\alpha_1, \dots, \alpha_{|\mathcal{A}_i|}\}$ be the pure strategies of player $i \in \mathcal{N}$; then

$$\mathbb{E}[V_{i,n}] = \sum_{\alpha_{-i} \in \mathcal{A}_{-i}} (u_i(\alpha_1; \alpha_{-i}), \dots, u_i(\alpha_{|\mathcal{A}_i|}))\hat{X}_{-i,n} = v_i(\hat{X}_n) \tag{D.12}$$

where with $\hat{X}_{-i,n}$ we symbolize the joint probability distribution for all players except for the $i^{th}$ player.

2. We move on to the second part of this proposition.

$$\|U_{i,n}\|_* = \|V_{i,n} - v_i(\hat{X}_n)\|_* \tag{D.13}$$
$$\leq \|V_{i,n}\|_* + \|v_i(\hat{X}_n)\|_* \tag{D.14}$$
$$\leq \max_{\alpha \in \mathcal{A}} |u_i(\alpha)||\mathcal{A}_i|/\varepsilon_n + \max_{\alpha \in \mathcal{A}} |u_i(\alpha)| \tag{D.15}$$
$$= \mathcal{O}(1/\varepsilon_n) \tag{D.16}$$

3. Finally for the norm of the bias term, let again $\mathcal{A}_i = \{\alpha_1, \dots, \alpha_{|\mathcal{A}_i|}\}$ be the pure strategies of player $i \in \mathcal{N}$; then

$$\|b_{i,n}\|_* = \|v_i(\hat{X}_n) - v_i(X_n)\|_* \tag{D.17}$$
$$= \|(u_i(\alpha_1; \hat{X}_{-i,n}) - u_i(\alpha_1; X_{-i,n}), \dots, u_i(\alpha_{|\mathcal{A}_i|}; \hat{X}_{-i;n}) - u_i(\alpha_{|\mathcal{A}_i|}; X_{-i;n}))\|_* \tag{D.18}$$

It is sufficient to examine one of the elements of the vector $b_{i,n}$:

$$|u_i(\alpha_1; \hat{X}_{-i,n}) - u_i(\alpha_1; X_{-i,n})| \tag{D.19}$$
$$= |\sum_{\alpha_2 \in \mathcal{A}_2} \cdots \sum_{\alpha_N \in \mathcal{A}_N} (\hat{X}_{2\alpha_2,n} \dots \hat{X}_{N\alpha_N,n} - X_{2\alpha_2,n} \dots X_{N\alpha_N,n})u_i(\alpha_1, \dots, \alpha_N)| \tag{D.20}$$
$$\leq \sum_{\alpha_2 \in \mathcal{A}_2} \cdots \sum_{\alpha_N \in \mathcal{A}_N} |\hat{X}_{2\alpha_2,n} \dots \hat{X}_{N\alpha_N,n} - X_{2\alpha_2,n} \dots X_{N\alpha_N,n}||u_i(\alpha_1, \dots, \alpha_N)| \tag{D.21}$$
$$= \mathcal{O}(\varepsilon_n) \tag{D.22}$$

∎

In this section we provide different algorithms and feedback models which connect to our general algorithm (FTGL) and model described in Section 3.2. We first present a useful proposition in order to calculate the permitted parameters of the algorithm in order for assumption A3 to be satisfied.

**Proposition D.2.** *1. For all step sizes of the form $\gamma_n = \gamma/n^p$, with $p < 1$ and noise bounds $\sigma_n = \sigma n^r$ assumption A3 is satisfied if*

$$\frac{2}{q} - p + 2r < \beta(1 - p) \text{ for some } \beta < 1 \tag{D.23}$$

*Furthermore, it holds that*

$$1/q + r < 1/2 \tag{D.24}$$

*2. For all step-sizes of the form $\gamma_n = \gamma/n$ and $\sigma_n = \sigma n^r$, assumption A3 holds as long as*

$$1/q + r < 1/2 \tag{D.25}$$

*Proof.* 1. Since $\gamma_n = \gamma/n^p$ and $\sigma_n = \sigma n^r$, assumption A3 is restated as

$$\delta_n = \frac{\sum_{k=1}^n \gamma_k^{1+q/2}\sigma_k^q}{[\sum_{k=1}^n \gamma_k]^{1+\beta q/2}} \tag{D.26}$$

$$= C_q\left(\sum_{k=1}^n 1/k^p\right)^{-1-\beta q/2}\sum_{k=1}^n 1/k^{p(1+\frac{q}{2})}k^{rq} \tag{D.27}$$

$$\leq C_q' n^{(1-p)(-1-\frac{\beta q}{2})}n^{1-p(1+\frac{q}{2})+rq} \tag{D.28}$$

$$\leq C_q' n^{-1-\frac{\beta q}{2}+p+\frac{p\beta q}{2}+1-p-\frac{pq}{2}+rq} \tag{D.29}$$

$$\leq C_q' n^{-\frac{\beta q}{2}+\frac{p\beta q}{2}-\frac{pq}{2}+rq} \tag{D.30}$$

Thus $\delta_n$ is summable if the exponent of $n$ is less than $-1$:

$$-\frac{\beta q}{2} + \frac{p\beta q}{2} - \frac{pq}{2} + rq < -1 \tag{D.31}$$

$$\frac{2}{q} - p + 2r < \beta(1-p) \tag{D.32}$$

The second expression of the proposition can be derived if we only keep the variable $a$ in the RHS of the above inequality

$$\frac{2}{q} - p + 2r < \beta(1-p) \tag{D.33}$$

$$(\frac{2}{q} - p + 2r)/(1-p) < \beta < 1 \tag{D.34}$$

$$\frac{2}{q} - p + 2r < 1 - p \tag{D.35}$$

$$1/q + r < 1/2 \tag{D.36}$$

2. Let $\gamma_n = \gamma/n$ and $\sigma_n = \sigma n^r$, then for assumption A3 we have

$$\delta_n = \frac{\sum_{k=1}^n \gamma_k^{1+q/2}\sigma_k^q}{[\sum_{k=1}^n \gamma_k]^{1+\beta q/2}} \tag{D.37}$$

$$= C_q\frac{\sum_{k=1}^n \frac{1}{k^{1+q/2}}k^{rq}}{[\sum_{k=1}^n \frac{1}{k}]^{1+\beta q/2}} \tag{D.38}$$

$$\leq C_q'(\log(n+1))^{-1-\beta q/2}n^{1-1-q/2+rq} \tag{D.39}$$

$$\leq C_q'(\log(n+1))^{-1-\beta q/2}n^{-q/2+rq} \tag{D.40}$$

Since the sum $\sum_{n=1}^\infty 1/(\log^{1+\varepsilon}(n)n^{1+\varepsilon'})$ is finite for all $\varepsilon, \varepsilon' > 0$; assumption A3 is satisfied as long as

$$-q/2 + rq < -1 \Rightarrow 1/q + r < 1/2 \tag{D.41}$$

∎

**Model D.1** ((FTRL) & Full information)**.** In this case players have access to their full payoff vector $v(X_n)$ for each round $n = 1, 2, \ldots$ and thus $V_{i,n} = v_i(X_n)$ for all $i \in \mathcal{N}$. All of the assumptions A1-A3 are satisfied; indeed

- (A1): Trivially satisfied since $b_{i,n} = 0$.
- (A2): Trivially satisfied since $U_{i,n} = 0$.
- (A3): From Proposition D.2 is satisfied for all the step-sizes of the form $\gamma_n \propto 1/n^p$, $p \in [0, 1]$. §

**Model D.2** ((FTRL) & Noisy payoff feedback)**.** In this setting at each round $n = 1, 2, \ldots$ players have access to a perturbed version of their full payoff vector $v(X_n)$ with a zero-mean noise $U_n$. Two examples of such noises that we consider in the experimental section are a zero-mean guassian noise and a uniform noise at $[-1.1]$. Both these noises satisfy (A2) with deterministic constant bounds for all $q \in [1, \infty]$. Thus

- (A1): Trivially satisfied since $b_{i,n} = 0$.
- (A2): Satisfied for all $q \in [1, \infty]$.
- (A3): From Proposition D.2 is satisfied for all the step-sizes of the form $\gamma_n \propto 1/n^p$, $p \in [0, 1]$. §

**Model D.3** ((FTRL) & Oracle-based feedback). Assume that each player chooses an action based on a given mixed strategy, and once every player has chosen an action, an oracle reveals to each player their corresponding pure payoff vector. Formally, at each round $n = 1, 2, \ldots$, each player chooses a pure strategy $\alpha_{i,n} \in \mathcal{A}_i$ based on a mixed strategy $X_{i,n} \in \mathcal{X}_i$ and subsequently observes the payoff vector

$$V_{i,n} = v_i(\alpha_n) = (u_i(\alpha_i; \alpha_{-i,n}))_{\alpha_i \in \mathcal{A}_i}. \tag{D.42}$$

Regarding our basic assumptions, we readily have $b_{i,n} = 0$ and $U_{i,n} = v_i(\alpha_n) - v_i(X_n)$; hence:

- (A1): Trivially satisfied since $b_{i,n} = 0$.
- (A2): Satisfied because $\|U_{i,n}\|_* = \|v_i(\alpha_n) - v_i(X_n)\|_* \leq 2 \max_{\alpha \in \mathcal{A}} \|v_i(\alpha)\|_*$, so $U_n$ has uniformly bounded moments for all $q \in [1, \infty]$.
- (A3): From Proposition D.2 is satisfied for all the step-sizes of the form $\gamma_n \propto 1/n^p$, $p \in [0, 1]$. §

**Model D.4** ((FTRL) & Payoff-based feedback). If the players only observe their realized rewards, they have to *construct* a model for $V_n$ based on incomplete information. This is the standard setting for multi-armed bandits, so it is often referred to as the "bandit feedback" model. In this case, the players' action selection process is as in Model D.3 above, but the feedback signal sequence $V_n$ is now reconstructed by means of the importance-weighted estimator

$$V_{i\alpha_i,n} = \frac{\mathbb{1}\{\alpha_{i,n} = \alpha_i\}}{\hat{X}_{i\alpha_{i,n}}} u_i(\alpha_n) \tag{IWE}$$

where $\hat{X}_{i,n} = (1 - \varepsilon_n)X_{i,n} + \varepsilon_n/|\mathcal{A}_i|$ is the mixed strategy of the $i$-th player at stage $n$. Compared to $X_{i,n}$ the player's actual sampling strategy is now recalibrated by an *explicit exploration* parameter $\varepsilon_n \to 0$ whose role is to stabilize the learning process. The idea behind this adjustment is that even if a strategy has zero probability to be chosen under $X_n$, it will still be sampled with positive probability thanks to the mixing factor $\varepsilon_n$.

The IWE estimator may be seen as a special case of the model (4) with $U_{i,n} = V_{i,n} - v_i(\hat{X}_n)$ and $b_{i,n} = v_i(\hat{X}_n) - v_i(X_n)$. All of the assumptions (A1)-(A3) are again satisfied; indeed:

- (A1): From Proposition D.1 $\|b_{i,n}\|_* = O(\varepsilon_n)$. Thus our assumption is satisfied since $\varepsilon_n \to 0$.
- (A2): Again from Proposition D.1 $\|V_{i,n} - v_i(\hat{X}_n)\|_* = O(1/\varepsilon_n)$ and thus the noise has bounded moments, $\sigma_n = \Theta(1/\varepsilon_n)$ for all $q \in [1, \infty]$.
- (A3): From Proposition D.2 is satisfied for all the step-sizes of the form $\gamma_n \propto 1/n^p$, $p \in [0, 1]$ and $\varepsilon_n \propto 1/n^r$, $r \in (0, 1/2)$.

§

**Model D.5** ((OptFTRL) & Full information). In this case the full payoff vector of each player is $V_{i,n} = v_i(\tilde{X}_n)$ for all $i \in \mathcal{N}$. As we proved above the state $\tilde{X}_n$ can be treated separately and thus

- (A1): Trivially satisfied since $b_{i,n} = 0$.
- (A2): Trivially satisfied since $U_{i,n} = 0$.
- (A3): From Proposition D.2 is satisfied for all the step-sizes of the form $\gamma_n \propto 1/n^p$, $p \in [0, 1]$. §

**Model D.6** ((OptFTRL) & Noisy payoff feedback). Again in this setting at each round $n = 1, 2, \ldots$ players have access to a perturbed version of their full payoff vector $v(\tilde{X}_n)$ with a zero-mean noise $U_n$. Two examples of such noises that we consider in the experimental section are a zero-mean guassian noise and a uniform noise at $[-1.1]$. Both these noises satisfy (A2) with deterministic constant bounds for all $q \in [1, \infty]$. Thus

- (A1): Trivially satisfied since $b_{i,n} = 0$.
- (A2): Satisfied for all $q \in [1, \infty]$.
- (A3): From Proposition D.2 and our specific analysis for (OptFTRL) is satisfied for all the step-sizes of the form $\gamma_n \propto 1/n^p$, $p \in [0, 1]$. §

**Model D.7** ((OptFTRL) & Oracle-based feedback). In this case the payoff signal $V_{i,n}$, which depends on the state $\tilde{X}_n$, is generated as follows: at each round $n = 1, 2, \ldots$, every player $i \in \mathcal{N}$ picks an action $\alpha_{i,n} \in \mathcal{A}_i$ based on $\tilde{X}_{i,n} \in \mathcal{X}_i$ and observes the pure payoff vector $v_i(\alpha_n) \equiv (u_i(\alpha_i; \alpha_{-i,n}))_{\alpha_i \in \mathcal{A}_i}$.

Each player's input signal is a special case of (4) with payoff feedback $V_{i,n} = v_i(\alpha_n)$, zero-mean noise $U_{i,n} = v_i(\alpha_n) - v_i(\tilde{X}_n)$ and bias $b_{i,n} = 0$ that satisfy all of the assumptions A1 - A3. In more detail, we have:

- (A1): trivially satisfied since $b_{i,n} = 0$.
- (A2): $\|U_{i,n}\|_* = \|v_i(\alpha_n) - v_i(\tilde{X}_n)\|_* \leq 2\max_{\alpha \in \mathcal{A}}\|v_i(\alpha)\|_*$ and thus the noise has bounded moments for all $q \in [1, \infty]$.
- (A3): From Proposition D.2 is satisfied for all the step-sizes of the form $\gamma_n \propto 1/n^p$, $p \in [0, 1]$. §

**Model D.8** ((OptFTRL) & Payoff-based feedback). As we mentioned in Model D.4, in this case players only observe their realized rewards; thus they have to *construct* a model for $V_n$ based on incomplete information. The players' action selection process is as in Model D.7 above, but the feedback signal sequence $V_n$ is now reconstructed by means of the importance-weighted estimator

$$V_{i\alpha_i,n} = \frac{\mathbb{1}\{\alpha_{i,n} = \alpha_i\}}{\hat{X}_{i\alpha_{i,n}}} u_i(\alpha_n) \tag{IWE}$$

where $\hat{X}_{i,n} = (1 - \varepsilon_n)\tilde{X}_{i,n} + \varepsilon_n/|\mathcal{A}_i|$ is the mixed strategy of the $i$-th player at stage $n$. Compared to $\tilde{X}_{i,n}$ the player's actual sampling strategy is now recalibrated by an *explicit exploration* parameter $\varepsilon_n \to 0$.

This type of feedback may be seen as a special case of the model (4) with $U_{i,n} = V_{i,n} - v_i(\hat{X}_n)$ and $b_{i,n} = v_i(\hat{X}_n) - v_i(X_n)$. All of the assumptions (A1)-(A3) are again satisfied; indeed:

- (A1): From Proposition D.1 $\|b_{i,n}\|_* = O(\varepsilon_n)$. Thus our assumption is satisfied since $\varepsilon_n \to 0$.
- (A2): Again from Proposition D.1 $\|V_{i,n} - v_i(\hat{X}_n)\|_* = O(1/\varepsilon_n)$ and thus the noise has bounded moments, $\sigma_n = \Theta(1/\varepsilon_n)$ for all $q \in [1, \infty]$.
- (A3): From Proposition D.2 is satisfied for all the step-sizes of the form $\gamma_n \propto 1/n^p$, $p \in [0, 1]$ and $\varepsilon_n \propto 1/n^r$, $r \in (0, 1/2)$. §

**Model D.9** ((MirrorProx) & Full information). In this case players have access to their full payoff vector $v(X_n)$ for each round $n = 1, 2, \ldots$; for the algorithm (MirrorProx) we observe two payoff vectors at each round and as stated in the specific analysis above, for each one of $v_i(X_{n+1/2})$ and $v_i(X_n)$, we have

- Assumption A1: Trivially satisfied since $b_{i,n} = 0$.
- (A2): Trivially satisfied since $U_{i,n} = 0$.
- (A3): From Proposition D.2 is satisfied for all the step-sizes of the form $\gamma_n \propto 1/n^p$, $p \in [0, 1]$. §

**Model D.10** ((MirrorProx) & Noisy payoff feedback). As before at each round $n = 1, 2, \ldots$ players have access to a perturbed version of their full payoff vector $v(X_n)$ with a zero-mean noise $U_n$. Two examples of such noises that we consider in the experimental section are a zero-mean guassian noise and a uniform noise at $[-1.1]$. Both these noises satisfy (A2) with deterministic constant bounds for all $q \in [1, \infty]$. Thus

- (A1): Trivially satisfied since $b_{i,n} = 0$.
- (A2): Satisfied for all $q \in [1, \infty]$.
- (A3): From Proposition D.2 and our specific analysis for (MirrorProx) is satisfied for all the step-sizes of the form $\gamma_n \propto 1/n^p$, $p \in [0, 1]$. §

We simply mention here that in the exact same way all of the assumptions (A1)-(A3) are satisfied for the second "intermediate" state of (MirrorProx).

**Model D.11** ((MirrorProx) & Oracle-based feedback). In this case, at each round $n$ each player $i \in \mathcal{N}$ chooses two pure strategies $\alpha_{i,n}$ and $\alpha_{i,n+1/2}$ successively based on the mixed strategies $X_{i,n}$, $X_{i,n+1/2}$ equivalently. Thus, the first payoff signal is $V_{i,n} = v_i(\alpha_n)$ with $b_{i,n} = 0$ and $U_{i,n} = v_i(\alpha_n) - v_i(X_n)$. Hence:

- (A1): Trivially satisfied since $b_{i,n} = 0$.
- (A2): Satisfied because $\|U_{i,n}\|_* = \|v_i(\alpha_n) - v_i(X_n)\|_* \leq 2\max_{\alpha \in \mathcal{A}}\|v_i(\alpha)\|_*$, so $U_n$ has uniformly bounded moments for all $q \in [1, \infty]$.
- (A3): From Proposition D.2 is satisfied for all the step-sizes of the form $\gamma_n \propto 1/n^p$, $p \in [0, 1]$, by also taking into account our specific analysis for (MirrorProx) presented above. §

The second payoff signal is $V_{i,n+1/2} = v_i(\alpha_{n+1/2})$ with $b_{i,n+1/2} = 0$ and $U_{i,n+1/2} = v_i(\alpha_{n+1/2}) - v_i(X_{n+1/2})$

- (A1): Trivially satisfied since $b_{i,n+1/2} = 0$.
- (A2): Satisfied because $\|U_{i,n+1/2}\|_* = \|v_i(\alpha_{n+1/2}) - v_i(X_{n+1/2})\|_* \leq 2\max_{\alpha \in \mathcal{A}}\|v_i(\alpha)\|_*$, so $U_n$ has uniformly bounded moments for all $q \in [1, \infty]$.
- (A3): From Proposition D.2 is satisfied for all the step-sizes of the form $\gamma_n \propto 1/n^p$, $p \in [0, 1]$, by also taking into account our specific analysis for (MirrorProx) presented above. §

**Model D.12** ((MirrorProx) & Payoff-based feedback). In this case, as we have already mentioned, players only observe their realized rewards and the feedback signal sequence $V_n$ is now reconstructed by means of the importance-weighted estimator

$$V_{i\alpha_i,n} = \frac{\mathbb{1}\{\alpha_{i,n} = \alpha_i\}}{\hat{X}_{i\alpha_i,n}} u_i(\alpha_n) \tag{IWE}$$

where $\hat{X}_{i,n} = (1 - \varepsilon_n)X_{i,n+1/2} + \varepsilon_n/|\mathcal{A}_i|$ is the mixed strategy of the $i$-th player at stage $n$, with $\varepsilon_n \to 0$.

The IWE estimator may be seen as a special case of the model (4) with $U_{i,n} = V_{i,n} - v_i(\hat{X}_n)$ and $b_{i,n} = v_i(\hat{X}_n) - v_i(X_n)$. All of the assumptions (A1)-(A3) are again satisfied; indeed:

- (A1): From Proposition D.1 $\|b_{i,n}\|_* = O(\varepsilon_n)$. Thus our assumption is satisfied since $\varepsilon_n \to 0$.
- (A2): Again from Proposition D.1 $\|V_{i,n} - v_i(\hat{X}_n)\|_* = O(1/\varepsilon_n)$ and thus the noise has bounded moments, $\sigma_n = \Theta(1/\varepsilon_n)$ for all $q \in [1, \infty]$.
- (A3): From Proposition D.2 is satisfied for all the step-sizes of the form $\gamma_n \propto 1/n^p$, $p \in [0, 1]$ and $\varepsilon_n \propto 1/n^r$, $r \in (0, 1/2)$.

## E    Experiments

We start this section by explaining in detail the two main games that our experiments are conducted.

### E.1.  Games.

1. In the archetypal game of *Battle of the Sexes*, a couple argues over which music to listen over the weekend. Both know that they want to spend the weekend together, but they cannot agree over what to do. The partner (A) prefers to audit a *Rock* band concert, whereas the partner (B) prefers a *Pop* music show. This is a classical example of a coordination game, analysed in game theory for its applications in many fields, such as business management or military operations. For the interested reader, check [26]. Since the couple wants to spend time together, if they go separate ways, they will receive no utility (set of payoffs will be $0, 0$). If they go either to a *Rock* or a *Pop* musical, both will receive some utility from the fact that they're together, but one of them will actually enjoy the activity. The description of this game in strategic form is therefore as follows:

Battle of Sexes

|       | *Rock*  | *Pop*   |
|-------|---------|---------|
| *Rock* | $(2, 1)$ | $(0, 0)$ |
| *Pop*  | $(0, 0)$ | $(1, 2)$ |

**Figure 3:** Equilibrium Structure: This game has two strict Nash equilibria, one where both go to the *Rock* concert, and another where both go to the *Pop* concert. There is also a mixed Nash equilibrium, where the players go to their preferred event more often than the other. For the described payoffs, each player attends their preferred event with probability $3/5$.

2. In the selfish routing game of *Pigou's Congestion Network*, we consider the simple network shown in Fig. 4. Two disjoint edges/paths connect a source vertex $O$ to a destination vertex $D$. Each edge is labeled with a cost function, which describes the cost (e.g., travel time) incurred by users of the edge, as a function of the amount of traffic routed on the edge. In the atomic version of the game the population of the drivers that uses a specific edge is an integer $x \in \{0, \cdots, N\}$. The upper edge has the constant latency function $\ell_1(x) = 1$, and thus it represents a route that is relatively long but immune to congestion. In the linear latency setting, the cost of the lower edge, which is governed by the function $\ell_2(x) = x/N$, increases as the edge gets more congested. In particular, the lower edge is cheaper than the upper edge if and only if less than $N$ drivers uses it.

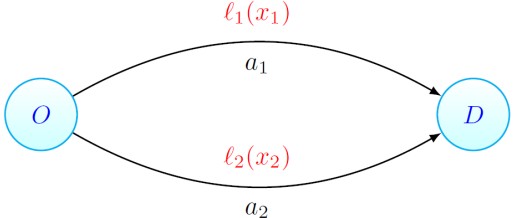

**Figure 4:** Pigou's Network

**E.2. Experimental setup and methodology.** Below, we will present separately the three archetypal instantiations of (FTGL) that we discussed in Appendix D, namely (FTRL),(OptFTRL) and (MirrorProx). All algorithms were run on *a*) a game of the Battle of the Sexes; and *b*) Pigou's linear version with $N = 1000$ atomic drivers. For each algorithm and each model we will present the performance of two well-studied regularizers: • entropic : $\theta_\alpha(x) = x_\alpha \log x_\alpha$ • euclidean : $\theta_\alpha(x) = x_\alpha^2/2$.

We will group our models with the following way: The first collection of figures for each algorithmic subsection will include the {oracle-based,payoff based/bandit} feedback model for the two aforementioned games for constant step-size and inverse-polynomial $\gamma_n \propto 1/n^{1/2}$. The latter one will present the {perfect,uniform-noise,gaussian-noise} feedback. Finally, the shaded areas around the curves represent the error bars in the execution for different random initializations.

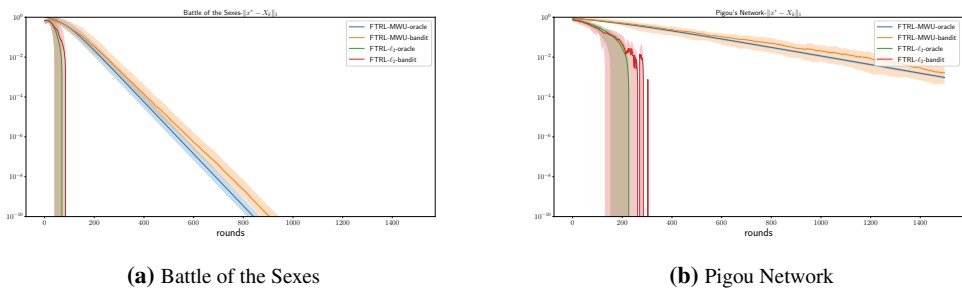

**(a)** Battle of the Sexes

**(b)** Pigou Network

**Figure 5:** FTRL: oracle-based, bandit; $\gamma_n = 0.05$

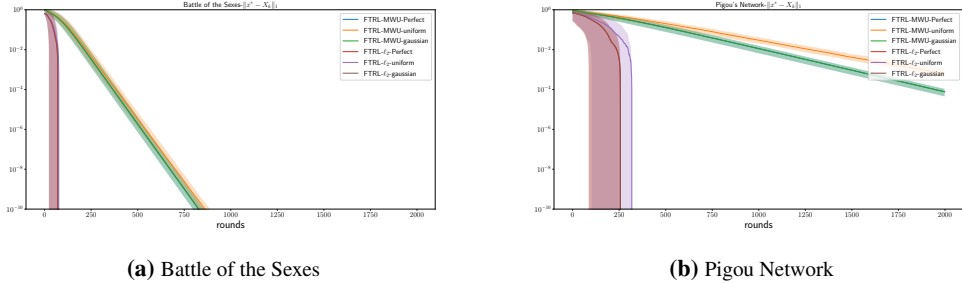

**(a)** Battle of the Sexes

**(b)** Pigou Network

**Figure 6:** FTRL: uniform, gaussian; $\gamma_n = 0.05$.

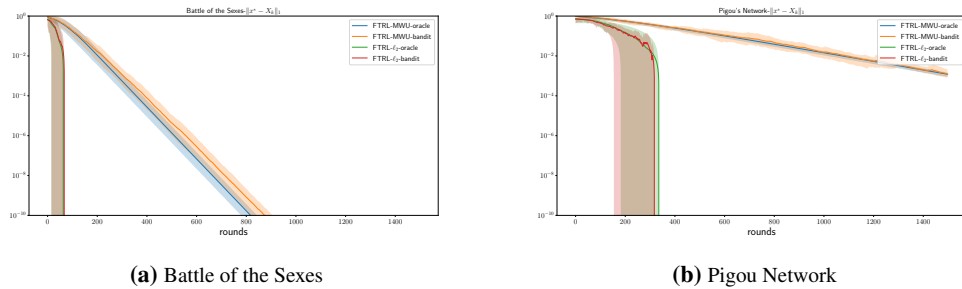

**(a)** Battle of the Sexes

**(b)** Pigou Network

**Figure 7:** FTRL oracle, bandit; $\gamma_n \propto 1/n^{1/2}$

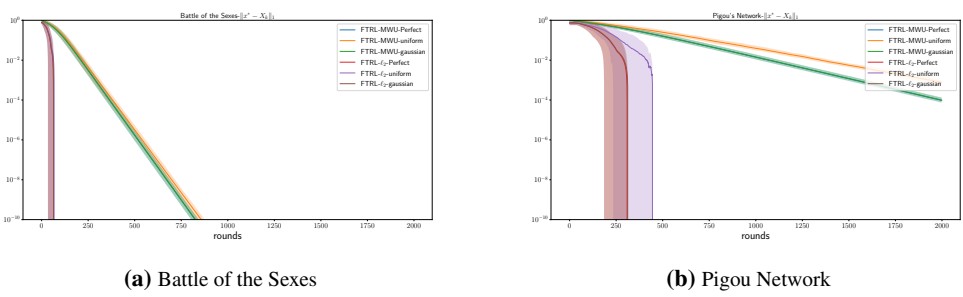

**(a)** Battle of the Sexes

**(b)** Pigou Network

**Figure 8:** FTRL: uniform, gaussian; $\gamma_n \propto 1/n^{1/2}$

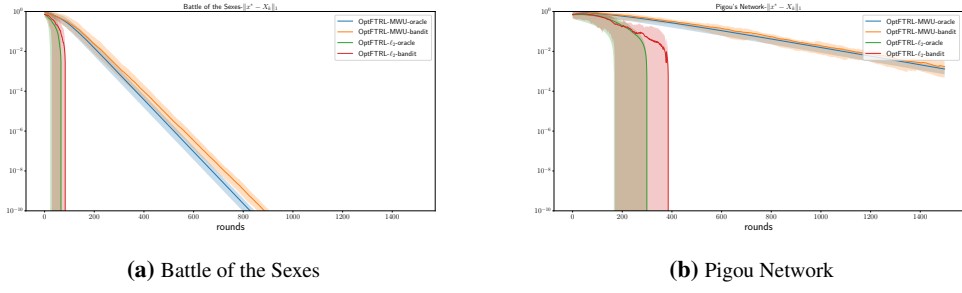

**(a)** Battle of the Sexes

**(b)** Pigou Network

**Figure 9:** OptFTRL: oracle-based, bandit; $\gamma_n = 0.05$

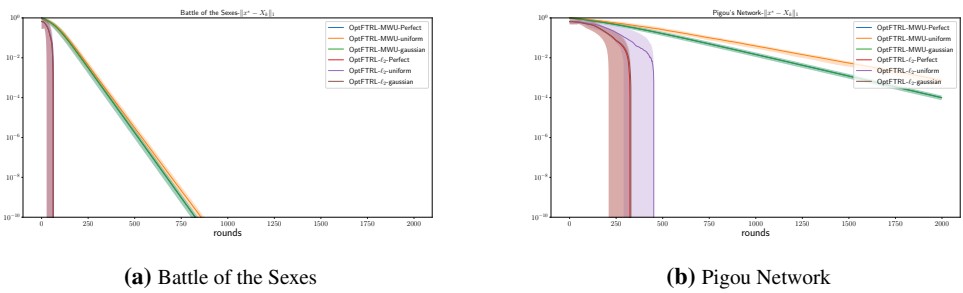

**(a)** Battle of the Sexes

**(b)** Pigou Network

**Figure 10:** OptFTRL: uniform, gaussian; $\gamma_n = 0.05$

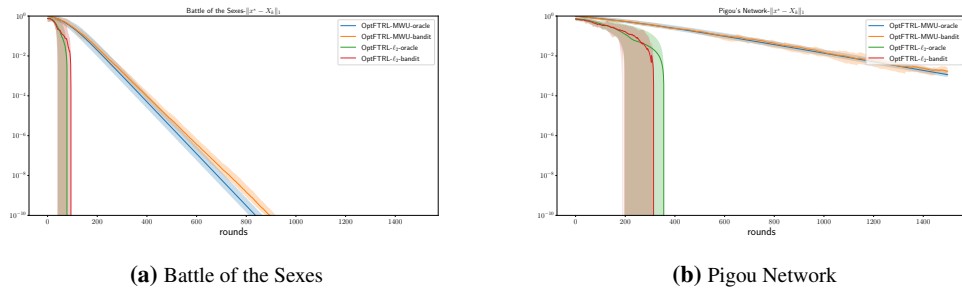

**(a)** Battle of the Sexes

**(b)** Pigou Network

**Figure 11:** OptFTRL: oracle-based, bandit; $\gamma_n \propto 1/n^{1/2}$

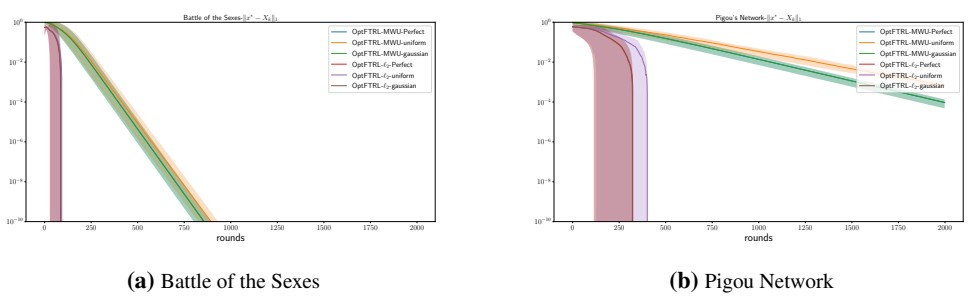

**(a)** Battle of the Sexes

**(b)** Pigou Network

**Figure 12:** OptFTRL: uniform, gaussian; $\gamma_n \propto 1/n^{1/2}$

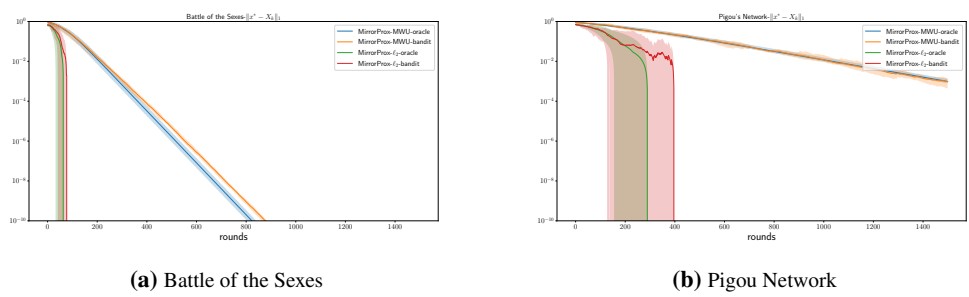

**(a)** Battle of the Sexes

**(b)** Pigou Network

**Figure 13:** MP: oracle-based, bandit; $\gamma_n = 0.05$

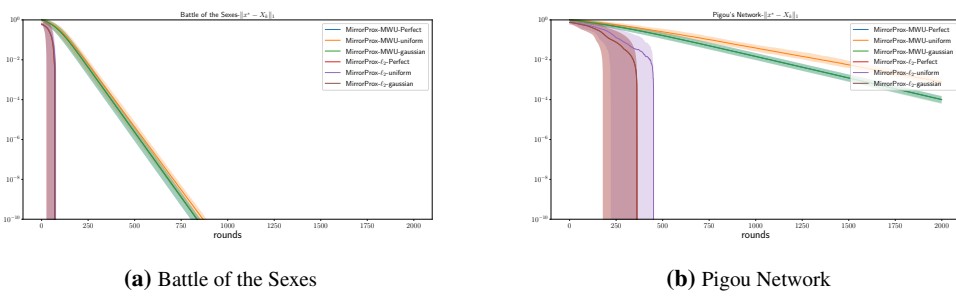

**(a)** Battle of the Sexes

**(b)** Pigou Network

**Figure 14:** MP: uniform, gaussian; $\gamma_n = 0.05$

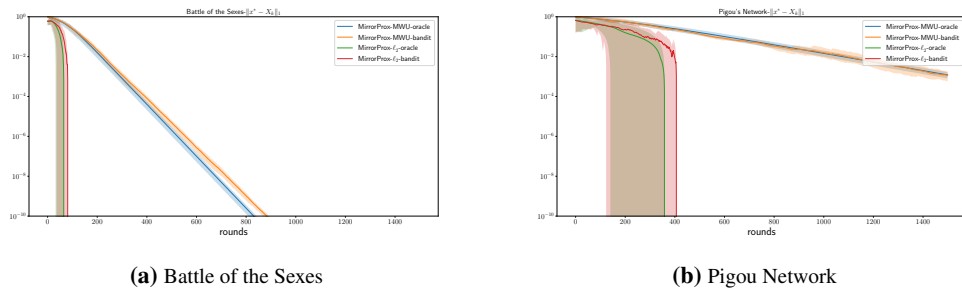

**(a)** Battle of the Sexes       **(b)** Pigou Network

**Figure 15:** mirror-prox (MP): oracle-based, bandit; $\gamma_n \propto 1/n^{1/2}$

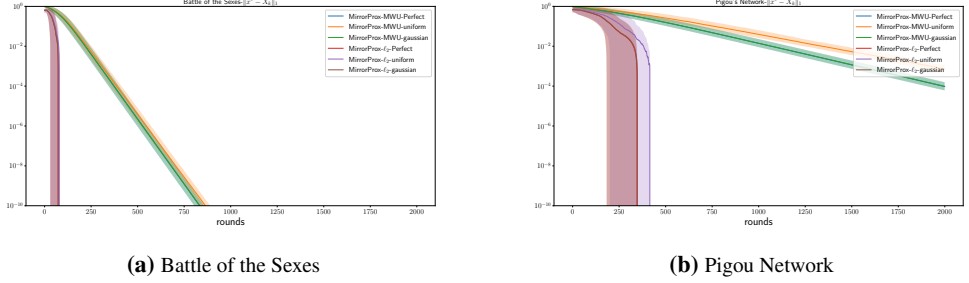

**(a)** Battle of the Sexes       **(b)** Pigou Network

**Figure 16:** MP: uniform, gaussian; $\gamma_n \propto 1/n^{1/2}$