# OpenReview forum: "On the Rate of Convergence of Regularized Learning in Games: From Bandits and Uncertainty to Optimism and Beyond"
_NeurIPS.cc/2021/Conference — NeurIPS 2021 Poster_

### Official Review · Reviewer_4S9g · 2021-07-16

**Rating:** 6
**Confidence:** 3

**Summary:**

This paper focused on normal-form games and proposed an algorithm framework called "follow the generalized leader" (FTGL). The framework covers the well-known "follow the regularized leader" methods as well as many other variants. The authors established FTGL's local convergence to strict Nash equilibria and further demonstrated that MWU converged linearly, while Euclidean projection methods converge within finite steps.

**Limitations And Societal Impact:**

See main review

**Main Review:**

This paper focused on normal-form games and proposed an algorithm framework called "follow the generalized leader" (FTGL). The framework covers the well-known "follow the regularized leader" methods as well as many other variants. In particular, the FTGL method updates a score vector with the payoff observed in every iteration and the strategy is induced by the score vector via a choice map function. This include well-known MWU method as a special case.

The main result states that as long as the score vector lies in a proper initialization zone, FTGL methods are guaranteed to converge. The convergence rate depends on the choice of step size as well as the choice map. In particular, MWU with constant step size converges linearly and Euclidean projection methods converges within finite steps.

While the results appear to be new, the fast convergence does not seem to be surprising under the assumption of strict Nash equilibrium and initialization condition. Actually the assumptions may have revealed extra information: as long as $\\|X_i - x_i^*\\|_\{1} < 1$, one can immediately recover $x_i^\*$ by simply picking the action with largest probability in $X_i$. Therefore, providing some discussion on how large  $W_\{init}$ is would make the result more solid. Showing $W_\{init}$ is unbounded alone doesn't necessarily leads to larger $\\{X_\{i, 1} = Q_i(Y_\{i,1}), Y_1 \in Q_\{init}\\}$, as scaling (or shifting, depending on the choice map) the score vector leads to exactly the same strategy.

Typos:

line 246, "thatthat";

line 248, "consdier"

**Time Spent Reviewing:**

2

---

> ### Author Response · Authors · 2021-08-10
> **Response to Reviewer 4S9g**
>
> Thanks for your time and your remarks. We reply to your two main points below:
>
> ### 1. **On picking the action with the highest probability.**
>
> The method you propose - each player playing $\arg\max_{a_i} x_{ia_i}$ would yield convergence to a vertex of the simplex within a single iteration. This would be ideal if players had knowledge of the underlying game, i.e., if they knew that the specific vertex is indeed a Nash equilibrium. In any other case, players could end up at any vertex of the simplex, which  could actually be catastrophic because not all vertices are Nash equilibria (strict or otherwise).  Hence, this procedure could actually lead players to play highly suboptimal strategies – e.g., strategy profiles where all players play a strictly dominated strategy.
>
> The importance of studying FTRL-type dynamics and algorithms lies precisely in that spurious convergence phenomena of this type cannot occur – see for example the recent reference [15] (numbering as in the paper). In fact, FTRL algorithms can only converge to strict Nash equilibria, independently of the players’ knowledge of the underlying game, hence the interest in completing the picture and studying the algorithm’s rate of convergence when it enters a neighborhood of a strict equilibrium.
>
>
> ### 2. **On the size of the initial neighborhood.**
>
> Roughly speaking, the basin of attraction of a given equilibrium is determined by **Eq. (9)** which introduces the set
>
> $\mathcal{B} = \begin{Bmatrix}x : v_{ia_i^*}(x) - v_{ia_i}(x) > 0\end{Bmatrix}$
>
> The actual initialization neighborhood needs to be slightly smaller than $\mathcal{B}$, depending on the value of $\delta$ (this is detailed in **Appendix C**, cf. **Lemma C.1** and the ensuing discussion). We will of course be happy to bring the relevant parts of **Appendix C** to the main body of the paper.
>
> Related to the above, the reviewer mentions that shifts by a constant may render unboundedness spurious as a qualification. [Incidentally, the regularized best response map of **Eq. (3)** is shift-invariant, not scale-invariant.] However, as can be seen from the definition of $W_M$ in **Eq. (10)**, the shift invariance has already been taken into account: the sets $W_M$ are defined in terms of score differences, so shifts by a constant do not impact the definition. Since $W_{\textrm{init}}$ is of the form $W_M$ for a suitably chosen $M$, unboundedness of $W_{\textrm{init}}$ does not refer to shifts either; specifically, $W_{\textrm{init}}$ remains unbounded even after taking the quotient by the equivalence relation $y_1 \sim y_2$ if $y_2$ is a constant shift of $y_1$. In particular, the smaller $M$ is, the larger both $W_M$ and $Q(W_{M})$ are.
>
> At any rate, we understand that the reviewer would prefer a clearer discussion of the basin of attraction of a strict equilibrium before **Theorem 1**, and not afterwards, as part of the proof. In this regard, we will be happy to incorporate the relevant parts of the above discussion (including the definition of $\mathcal{B}$ and the like) in the statement of **Theorem 1**, so as to give the reader a clearer idea of the size of the basin of attraction.
>
> We trust and hope that the above alleviates your concerns, and we look forward to engaging in an open-minded discussion if you have any more questions about our paper.
>
> Kind regards,
>
> *The authors*

---

> > ### Comment · Reviewer_4S9g · 2021-08-26
> > **Re: Response to Reviewer 4S9g**
> >
> > Thanks for the detailed explanation. The rebuttal addressed my concerns and I have raised my score accordingly.

---

### Official Review · Reviewer_FAhb · 2021-07-17

**Rating:** 7
**Confidence:** 3

**Summary:**

# Summary

The paper investigates the important issue of convergence in games.
A local convergence result is shown in N player general sum games, where local
means within a neighborhood of a strict Nash equilibria.
Convergence is attained with the proposed follow the generalized leader (FTGL)
framework, resembling the classical follow the regularized leader method
from online learning. Futhermore convergence is shown under a set of assumptions
that includes various forms of feedback, from deterministic to bandit feedback.
Connections with FTGL and other well-known algorithms are also made.


**Limitations And Societal Impact:**

Limitations are discussed throughout the paper. There is not discussion of potential negative societal impact.

**Main Review:**


# Main Review

## Summary of Review

Overall the paper is well written with significant contributions to the field of learning in games.
In particular, although only local convergence to a strict Nash equilibrium is considered,
a convergence result for a large family of algorithms such as FTGL in general sum games is likely to be impactful.
Furthermore, having a unified framework for studying various forms feedback (from deterministic to unbiased to bandit) and showing convergence is of great value.
I would expect that similar techniques might be useful for examining stronger convergence results if more assumptions are made on the game such as two-player zero-sum.

I do however have some issues regarding the discussion of related work and the claimed connections of FTGL with other well-known algorithms (see clarity section below).


## Originality

I believe the results to be novel see summary above.

## Quality

The paper is complete in its investigation of local convergence for the FTGL method
and results are supported with extensive proofs, using well-known tools from probability theory and convex analysis.
I would like to see more clarification of the fixed stepsize result referred to in lines 295-298. It is difficult to see where this result is located in Appendix C, it would be better to include a more exact pointer.
This result is important because other related works show convergence for a fixed stepsize [2].

## Clarity
I believe the paper to be well-written overall, however, improvements can be made with respect to discussion of related work, and connections between FTGL and other methods.

### Discussion of related work

Comparisons of related work are made in the related work section and throughout the paper.
There is an emphasis on stohcastic vs determinstic (See footnote 2 for example). It is claimed that a lot the related works depend on deterministic feedback for convergence as opposed to this paper studying a more general feedback model for which results from the related work do not apply (expect the works of [1] for example).
I too believe this statement to be true, however, I believe an important main driver of convergence in a lot of the works cited in these cases like those in footnote 2 or like [2], is not just the fact that it is determinstic but also because the games are monotone and Lipschitz smooth (don't need strongly monotone in the most general settings). But it should be mentioned that there is a significant important distinction in the quality of the results attained in these works, that they are global convergence guarantees.
Take the original extra gradient method from [3] for example, it is shown that the procedure will converge to an equilibrium without assuming starting in a neighbourhood of an equilibrium.
Or take for example an even closer related work [2], where convergence of the optimistic multiplicative weights algorithm (an example of optimistic follow the regularized leader) is shown to converge in zero-sum matrix games with unique equilibrium if a sufficiently small constant stepsize is used.
Therefore, I would suggest to emphasize that tho the paper investigates convergence in general sum games with possibly stochastic feedback the convergence results are local while a lot of the related works focus on global convergence with determistic feedback; except perhaps for the mentioned related work [1] where I believe global convergence is shown in potential games even with bandit feedback.

### Connections between FTGL and other methods

#### FTGL vs FTRL
Although FTGL resembles FTRL I believe it is different in a fundamental way by how the stepsizes are used. FTRL with a sequence stepsizes is generally understood to mean that at a each round when a stepsize is selected it is applied to all the past feedback vectors in the same way (see for example [4]). This is different that FTGL unless constant stepsizes are used.

#### FTGL and Mirror Prox or Mirror Descent

I believe that FTGL is more closely related to FTRL than mirror descent since it maintains an average in the dual space $Y_{i,n}$ and then projects back to the primal space when needed. This setup can sometimes be equivalent to mirror descent  (for example see [4]), but without further assumptions this approach is different
than mirror descent (projected gradient descent as an example) which maps the primal iterate to the dual space before taking
a gradient step. For example in the case of example 3.1 they are equivalent but in the case of example 3.2 they are not. Therefore, I do not see how the FTGL framework can capture mirror prox in general (which can be written as optimistic mirror descent [5]). I see there are discussions in the appendix to justify this connection, however, I do not believe mirror prox or extra gradient can be recovered if one averages in the dual space like FTGL.



## Significance
See summary above.

## Other minor suggested improvements

* I think there is a typo on line 134 $supp(x_i^\ast)$ should be $supp(x^\ast)$ ?
* What is the value of $q$ in assumption A3? is it the same $q$ as in A2?
* Why is it assumed that $h_i$ is decomposable? For example do you expect $h_i(x_i) = <x_i, M x_i>$ the matrix norm for an arbitrary positive definite matrix (hence $h_i$ is strongly convex) to also work? what is the difficulty when we cannot decompose $h_i$? A discussion would be nice.
* On remarkk 3.1. This remark seems to be closely related to the Legendre assumption that is commonly used when defining mirror descent in online learning, see for example [4]. This might be interesting to mention or investigate.
* On model 3 for OptFTRL. This perspective suggests that optimism is a nuissance to deal with since it contributes to bias. However, in online learning optimism improves regret bounds for predicatble problems and so helps with convergence [5,6].
A comment on these two different perspectives would be interesting.


References

1. Johanne Cohen, Amélie Héliou, and Panayotis Mertikopoulos. Learning with bandit feedback in potential games. In
NIPS ’17: Proceedings of the 31st International Conference on Neural Information Processing Systems, 2017.

2. Constantinos Daskalakis and Ioannis Panageas. Last-iterate convergence: Zero-sum games and constrained min-max
optimization. In ITCS ’19: Proceedings of the 10th Conference on Innovations in Theoretical Computer Science, 2019.

3. G. M. Korpelevich. The extragradient method for finding saddle points and other problems. Èkonom. i Mat. Metody, 12:
747–756, 1976.

4. Orabona, F., 2019. A modern introduction to online learning. arXiv preprint arXiv:1912.13213.

5. Rakhlin, S. and Sridharan, K., 2013. Optimization, Learning, and Games with Predictable Sequences. Advances in Neural Information Processing Systems, 26, pp.3066-3074.

6. Vasilis Syrgkanis, Alekh Agarwal, Haipeng Luo, and Robert E. Schapire. Fast convergence of regularized learning in
games. In NIPS ’15: Proceedings of the 29th International Conference on Neural Information Processing Systems, pages
2989–2997, 2015.

**Time Spent Reviewing:**

6

---

> ### Author Response · Authors · 2021-08-10
> **Response to Reviewer FAhb**
>
> Thank you very much for the detailed and thoughtful review, and for your positive evaluation and assessment. We reply to your precise questions below:
>
> ### 1. **On monotone games.**
>
> Thanks for this remark. We would first like to point out that there is a fundamental difference between learning in continuous games and finite games (the topic of our paper). The notion of monotonicity (and Lipschitz smoothness) is inherently tied to continuous games, so our objective was to make a clear separation between these two different branches of the literature to avoid undue comparisons.
>
> To elaborate on this, the mixed extension of a finite game can of course be seen as a continuous game played over a product of simplices and multilinear payoff functions. However, because of this multilinearity, mixed extensions of finite games are very rarely monotone. In fact, the only relevant class of finite games whose mixed extension is monotone is the class of two-player, zero-sum games (and this, precisely because these games are *bilinear*). Beyond this, even finite potential games are generically not monotone – except in trivial cases, e.g., if the game is constant. More to the point, finite games generally admit several isolated equilibria; by contrast, monotone games *never* admit *isolated* equilibria (and strictly/strongly monotone games only admit a unique equilibrium).
>
> Essentially, the only point of intersection between these two regimes is the class of two-player, zero-sum games – but in this case, last-iterate convergence relies crucially on having access to full, perfect payoff information. Specifically, the *extra-gradient algorithm of Korpelevich*, the *optimistic variant of Popov & Rakhlin–Sridharan*, and the *optimistic MWU algorithm of Daskalakis and Panageas* all require perfect, deterministic feedback to achieve convergence. By contrast, if the players only have access to noisy payoff feedback, the recent works of *Chavdarova et al.* (NeurIPS 2019) and *Hsieh et al.* (NeurIPS 2020) showed that extra-gradient / optimistic methods fail to converge to a Nash equilibrium in two-player zero-sum games, much like their non-optimistic counterparts.
>
> We state the above to highlight the fact that equilibrium convergence results for monotone games are of a fundamentally different nature compared to results for finite games: the former are usually *global* in nature (since monotonicity is a global structural assumption), while the latter are necessarily local in nature. It is for this reason that we did not include an extended discussion of convergence results for monotone games, and why we focused on the deterministic / stochastic separation.
>
> Of course, we will be more than happy to bring a version of the above discussion and the pointers provided in the paper – thanks again for your remark.
>
>
>
> ### 2. **On the relation between FT{R,G}L, Mirror-Prox and Mirror Descent.**
>
> The reviewer is touching on an issue where the literature is somewhat cloudy in terms of nomenclature. There are indeed several closely related methods with different names and, regrettably, these names are sometimes used interchangeably in the literature; the main points to keep in mind are as follows:
>
> * *The role of the step-size*: a first difference is whether payoff vectors (or gradients) enter the algorithm with different or equal weights. In the notation of our paper, this corresponds to taking $Y_t = \sum_s \gamma_s V_s$ versus $Y_t = \gamma_t \sum_s V_s$: in the former case, the parameter $\gamma_t$ plays the role of a step-size; in the latter, it plays the role of an “averaging” or “regularization” parameter. The first case corresponds to mirror descent, whereas the latter corresponds to dual averaging (Nesterov, 2009; Xiao, 2010). Obviously, the two schemes coincide when $\gamma$ is constant, and because FTRL was initially analyzed with a *constant parameter* (see e.g., Shalev-Shwartz, 2011), there is no consensus in the literature whether FTRL should be seen as a version of mirror descent or of dual averaging. To clear up any confusion, our paper takes the “mirror descent” interpretation of FTRL: *specifically, payoff signals enter the algorithm with (potentially) different weights.*
>
> * *The primal-dual interplay*: a second difference is whether the aggregation of payoff vectors (or gradients) occurs solely in the dual space, or if a method takes a Bregman proximal step directly from a primal iterate. In the context of mirror descent, this difference is sometimes referred to as *lazy* (the former) vs. *eager* (the latter), cf. Shalev-Shwartz (2011) and Zinkevich (2003). [In regards to mirror-prox, this gives rise to the difference between Nemirovski’s Mirror-Prox algorithm and Nesterov’s Dual Extrapolation variant (which, however, incorporates elements of both Mirror-Prox and Dual Averaging).]
>
> Importantly however, this difference only arises if the regularizer is non-steep; in the steep case, both algorithms are equivalent (Zhou et al., SIOPT 2020). Therefore, the algorithm provided is exactly Mirror-Prox in the steep or unconstrained case (where lazy = eager), and it should perhaps be called “lazy Mirror-Prox” in the non-steep, constrained case. However, we did not wish to go into a lengthy detour into the intricacies of Bregman proximal methods (and/or their single-query variants, where there is even more confusion in the literature), so we opted for the simpler name.
>
> We hope that the above clarifies our choices of terminology – and we would of course be happy to explain this in the paper as well.
>
>
>
> ### 3. **Constant step-size for Optimistic FTRL**
> We thank the reviewer for pinpointing the confusion regarding the constant step-size for Optimistic FTRL. Indeed there is a typo in line 295; the discussion and proof of this claim are elaborated in Appendix D (lines 671-685), not Appendix C.
>
> *More specifically, for both the cases of full information and oracle-based feedback, the exclusion of constant step-size is due to the bias term $||b_n|| = \mathcal{O}(\gamma_n)$ which implies that $\gamma_n$ needs to go uniformly to $0$ ( for Assumption 1 to be satisfied). However, in Appendix D by taking into account the fact that Optimistic FTRL requires two initializations we prove that no bias term is introduced and the constant step-size is indeed feasible. Notice that in the bandit case no such problem emerges, since the bias term is $||b_n|| = O(\epsilon_n)$ which goes uniformly to zero if $\epsilon_n$ does so.*
>
> ### 4. **Minor remarks.**
> * *Typos* : consider them fixed, many thanks for spotting them!
> * *On the value of $q$* : yes, it should be the same, we will clarify this.
> * *Decomposability of $h$* : this is done for ease of notation and presentation (the expression for the rates would be much more cumbersome with a non-decomposable $h$).
> * *The Legendre assumption* : absolutely correct, we will mention it.
> * *Optimism as a nuisance* : it is not a “nuisance” per se, but it doesn’t help in the presence of uncertainty, so it ends up contributing to the bias of the driver. The benefit can be seen in the deterministic case – but because we focus on learning under uncertainty, it is not as apparent. We will discuss this to make the distinction clear, thanks for pointing it out.
>
>
> Thank you again for your detailed comments and your positive evaluation!
>
> Kind regards,
>
> *The authors*

---

### Official Review · Reviewer_mbqJ · 2021-07-20

**Rating:** 6
**Confidence:** 3

**Summary:**

This paper proposes an abstraction of the online Follow-the-(Blank)-Leader type of methods they call Follow-the-Generalized-Leader (FTGL) that not only incorporates various regularization functions but also different feedback models. The authors provide a general convergence analysis for the family of methods over the problem of finding *strict* Nash equilibria in *finite* games. The main insight of the work is to claim that the convergence rate in this problem only depends on the regularization function used and *not* on the different feedback models, e.g. full information feedback vs bandit feedback.

**Limitations And Societal Impact:**

The societal impact was adequately addressed. I would like to see more discussion around the limitations of this work. The authors could answer questions such as how much this exact problem has been studied, what results were known for it, and what can be done for future work (or what should be investigated).

**Main Review:**

This submission was an interesting read given its somewhat surprising conclusions. It is clearly written and easy to follow. However, it seems like the authors aim to claim two contributions (this might not be true): (i) abstracting FTRL into FTGL and giving convergence analysis on a specific problem; (ii) new convergence results for the specific problem of strict Nash equilibria in finite games. I believe (i) is unnecessarily abstract and confusing for two reasons: first, we know how to do FTRL with different regularization functions (Bregman divergences as potential functions), so the fact that the regularization function is written more generally is not a contribution; second, it is not quite clear what the exact assumptions are w.r.t. different feedback models when they are all lumped in together into one.

More importantly, if the main contribution of this paper is showing new convergence results for finding strict Nash equilibria in finite games, then the authors should pick the specific algorithm(s) they use, prove and provide their convergence results, and draw conclusions. The unified framework might be appealing and seem elegant, but in this case, in my opinion, it doesn't contribute anything and obfuscates from the main problem at hand. In particular, the studied problem seems amenable to the Euclidean $l_2$ norm regularization given the geometry around the strict equilibria. The result suggests convergence to the desired points in a finite number of steps -- since the problem is solved in finite number of steps, for comparison one must look at the exact number of steps then instead of bringing in MW as another method to compare to. Now given $l_2$ regularization, is there any convergence rate difference between various feedback models, e.g. full information and bandit? I would be very surprised if not. And note that even if in your analysis you get identical rates, you cannot make any claims about these rates without any lower bounds -- maybe with your analysis they have the same rate but it's possible to do things much faster with full information. This same logic should apply to experiments, get rid of MW experiments, make the claim that there is convergence in finite number of steps, and compare the exact number for different methods with $l_2$ regularization.

Overall, the paper has interesting ideas, insights, and results, but, as presented, I don't think it's ready for publication. Weak reject.

**Time Spent Reviewing:**

8

---

> ### Author Response · Authors · 2021-08-10
> **Response to Reviewer mbqJ**
>
> We thank the reviewer for their time and remarks. Regrettably, there seems to be some misunderstandings regarding our paper’s contributions, which we hope to clarify in our point-to-point replies below.
>
> ### 1. **Regarding the abstraction from FTRL to FTGL and the unified framework.**
>
> The reviewer is factually mistaken when stating that we claim the use of a general regularizer as a contribution. FTRL has always been defined with a general Bregman function as a regularizer – this is well known and definitely not the point of our paper.
>
> Instead, the point of the abstraction from FTRL to FTGL is to provide a much more *flexible model for the sequence of payoff signals that each player best-responds to.* Owing to this abstraction, our analysis covers at the same time the **standard FTRL** algorithm of Shalev-Shwartz and Singer, the **Optimistic FTRL** algorithm of Rakhlin and Sridharan, the **Mirror-Prox / Dual Extrapolation** algorithms of Nemirovski and Nesterov, etc. Additionally, our feedback model allows us to analyze each of these algorithms, not only in the “*full information*” case (each player observes their mixed strategy payoff vector), but also in the “*oracle-based*” case (each player observes their *pure strategy* payoff vector), and even the “*bandit case*” (where each player only observes their realized payoff and has no other information about the game being played). These algorithms and settings have all come under intense scrutiny in recent years, so there is significant gain in including them in our analysis.
>
> To the best of our knowledge (and, by all accounts, also that of the reviewers’), there are no comparable convergence rate results in the literature, even for a subset of these algorithms. So, while it is true that our paper is stated at a level of abstraction that is more demanding than a paper studying a single algorithm with a single type of feedback, the benefit is that we are able to obtain tight convergence results for a broad range of algorithms under starkly different feedback assumptions. The unified framework we provide is therefore necessary to compare different algorithms and feedback assumptions – otherwise, all these algorithms would have to be studied in isolation, without any connecting tissue between them.
>
> We are aware that this level of generality can be demanding, so we consciously took the time to include several instructive models and examples for each of our assumptions (cf. the presentation of **Models 1–3** i pp. 5-6). This discussion was subsequently expanded on in **Appendix D** (**Models D.1–D.12**), where we showed exactly how each algorithm with each individual type of feedback can be recovered from our general model. We understand that the reviewer would prefer this discussion in the main text (to avoid “lumping”), and we would be happy to transfer it in our revision.
>
>
>
> ### 2. **On the sharpness of the derived bounds.**
>
> The bounds we provide are indeed sharp. To see this, consider a single-player game with two actions “$0$” and “$1$”, and payoffs $u(0) = 0$, $u(1) = 1$. Then, if e.g., FTRL is run with “full information” feedback, the probability $X_t$ that the player plays “$1$” at time $t$ is *exactly* equal to
>
> $X_t = 1 - \phi(c - \sum_{s=1}^{t} \gamma_s u(1) ) = 1 - \phi(c - \sum_{s=1}^{t} \gamma_s)$
>
> where $\phi$ is the rate function of Eq. (7) and $c$ is an initialization constant. This simple derivation shows that MWU converges to the game’s (strict) equilibrium at a rate of exactly $exp(-\Theta(\sum_{s=1}^{t} \gamma_s))$, whereas Euclidean methods achieve an equilibrium after a finite number of iterations – in particular, as soon as $\sum_{s=1}^{t} \gamma_s$ exceeds $c$. It thus follows that the rates provided by **Theorem 1** are, in general, unimprovable.
>
> The sharpness of the provided rates can also be seen in the extensive numerical experiments we present in **Fig. 1** and **Appendix E**. In particular, we would like to stress that the faster convergence rate of Euclidean algorithms is highly surprising and flies in the face of conventional wisdom. A regret-based reasoning would suggest the use of entropic regularization (which, ceteris paribus, has much better regret guarantees), but our analysis shows that a Euclidean regularizer is much more suitable for achieving convergence to equilibrium in a game-theoretic setting.
>
> It is for this reason that we insisted on the comparison between entropic and Euclidean regularization in the simulations: given the widespread popularity of the exponential weights algorithm, we believe this is an important take-away message for the community. [The Pigou network example of **Fig. 1b** is especially telling in this regard!]
>
> We will of course be happy to include a version of this discussion in our revision.
>
>
>
> ### 3. **On the exact number of iterations required in the Euclidean case.**
>
> Regarding the exact number of iterations per realization, there is indeed a difference between full information and bandit feedback. This is already seen in Fig. 1 (and in more detail in **Appendix E**), and we will be happy to provide a Euclidean-specific, “zoomed in” version of our experiments to illustrate this realization-based difference.
>
> [As a side remark, we would also like to point out here that, in the steep case, the full-information and bandit cases exhibit the same asymptotic rates in terms of $t$ – see e.g., the slope of the MWU plots in Fig. 1.]
>
> We trust and hope that the above clarifies any misunderstandings about our paper’s contributions. Thank you again for your time and input, and we look forward to a constructive discussion if you have any further questions.
>
> Kind regards,
>
> *The authors*

---

> ### Comment · Reviewer_mbqJ · 2021-09-04
> **Post Rebuttal**
>
> Given the author rebuttal and the discussion with the other reviewers, I am raising my score to 6. Contrary to my belief after my first read of the paper, I was persuaded that this work has more than sufficient significance for the experts in this area and its presentation is not flawed in major ways.

---

### Decision · Program_Chairs · 2021-09-27

**Decision:**

Accept (Poster)

**Comment:**

After discussions, reviewers agree that this paper makes solid contribution for
advancing the understanding of the convergence rate for regularized learning in games.
Please do incorporate all the suggestions/discussions from the reviews into the final version.